# De-AntiFake: Rethinking the Protective Perturbations Against Voice Cloning Attacks

Wei Fan [1]  Kejiang Chen [1]  Chang Liu [1]  Weiming Zhang [1]  Nenghai Yu [1]

## Abstract

The rapid advancement of speech generation models has heightened privacy and security concerns related to voice cloning (VC). Recent studies have investigated disrupting unauthorized voice cloning by introducing adversarial perturbations. However, determined attackers can mitigate these protective perturbations and successfully execute VC. In this study, we conduct the first systematic evaluation of these protective perturbations against VC under realistic threat models that include perturbation purification. Our findings reveal that while existing purification methods can neutralize a considerable portion of the protective perturbations, they still lead to distortions in the feature space of VC models, which degrades the performance of VC. From this perspective, we propose a novel two-stage purification method: (1) Purify the perturbed speech; (2) Refine it using phoneme guidance to align it with the clean speech distribution. Experimental results demonstrate that our method outperforms state-of-the-art purification methods in disrupting VC defenses. Our study reveals the limitations of adversarial perturbation-based VC defenses and underscores the urgent need for more robust solutions to mitigate the security and privacy risks posed by VC. The code and audio samples are available at https://de-antifake.github.io.

## 1. Introduction

With rapid advancements in Voice Cloning (VC) technology, generating highly realistic speech from just a few seconds of a target speaker's voice is now possible. This technology has diverse applications, including enhancing virtual assistants,

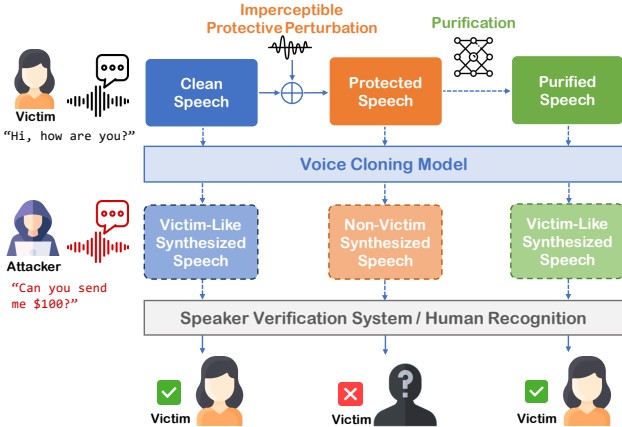

*Figure 1.* Illustrations of existing protection methods against voice cloning attacks in the presence of adversarial purification. The dashed lines represent the operations of the attacker.

chatbots, and providing assistive devices for individuals with speech impairments (Koronka, 2024; OpenAI, 2024). However, these advancements also pose risks for illegal uses, such as deceiving individuals, bypassing speaker verification systems, or violating copyrights. For instance, VC technology has been used to deceive identity verification systems in banks and government agencies (Nick Evershed, 2023; Cox, 2023). In a separate incident, scammers impersonated a CFO during a fraudulent call, tricking an employee into transferring $25 million (Andrews, 2024). These incidents highlight the potential of VC to deceive both digital systems and humans, raising significant security and privacy concerns. In response, organizations such as OpenAI and the FTC have released reports on VC's implications (OpenAI, 2024; FTC, 2024).

To address emerging threats posed by VC, researchers have explored various solutions, including proactive and passive detection, as well as proactive defense mechanisms (Wenger et al., 2021; Liu et al., 2024b; San Roman et al., 2024; Chen et al., 2024; Liu et al., 2024a; Ren et al., 2023; Deng et al., 2023; Ji et al., 2024; Li et al., 2024; Blue et al., 2022).

Among these approaches, protective perturbations are regarded as a promising technique for safeguarding speech. By adding imperceptible distortions to speech, they prevent

[1]Anhui Province Key Laboratory of Digital Security, University of Science and Technology of China, Hefei, China. Correspondence to: Kejiang Chen <chenkj@ustc.edu.cn>.

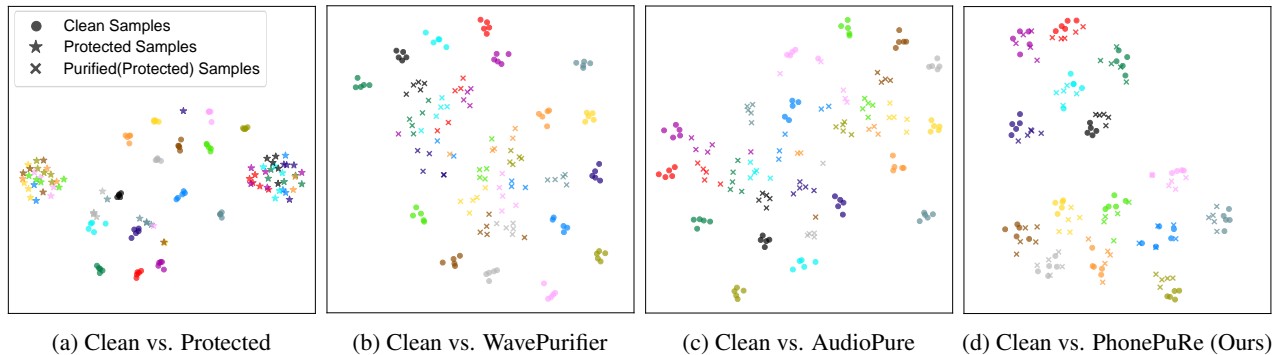

(a) Clean vs. Protected  (b) Clean vs. WavePurifier  (c) Clean vs. AudioPure  (d) Clean vs. PhonePuRe (Ours)

*Figure 2.* Comparison of sample distributions (Clean vs. Protected/Purified) in the VC model embedding space. Different colors represent different speakers. (a) Protected samples. (b-c), purified samples obtained by existing methods (Guo et al., 2024; Wu et al., 2023), which introduce distortions in the embedding space, including ❶ reduced interclass separability and ❷ deviation of purified samples from their clean counterparts. (d) Our method aligns purified samples more closely with their original clean versions.

VC models from accurately replicating authentic features of a speaker (Yu et al., 2023; Huang et al., 2021; Liu et al., 2023; Yang et al., 2024; Li et al., 2023b; Wang et al., 2023; Dong et al., 2024), achieving impressive results in reducing the effectiveness of VC attacks. For example, applying VC to protected speech results in synthesized speech with both lower speaker verification accuracy and a perceptually distinct timbre.

While protective perturbation methods can prevent VC models from deceiving both machines and humans under ideal conditions, their performance remains uncertain in more realistic scenarios where attackers may employ purification methods to eliminate the perturbations used for defense prior to performing VC. If these methods cannot remain effective against such purification strategies, they risk giving users a false sense of security. Therefore, it is essential to systematically evaluate their effectiveness in more realistic contexts that include potential purification methods, as shown in Figure 1.

In this paper, we perform comprehensive evaluation of protective perturbation methods against VC under a realistic threat model involving potential purification methods, with extensive experiments on the most common perturbation techniques and state-of-the-art VC models. Our results indicate that these protection methods are vulnerable to existing adversarial purification techniques. To the best of our knowledge, this is the first attempt to explore the vulnerabilities of protective perturbation-based VC defenses.

Moreover, we observe that existing purification methods introduce systematic distortions in VC model embedding spaces, as shown in Figure 2b and Figure 2c. Since VC models rely on fine-grained feature information to replicate speaker voices accurately, such distortions degrade their performance in VC tasks. Building on this insight, we propose a novel two-stage adversarial purification method that reduces

embedding inconsistencies caused by current methods: (1) In the first stage, we utilize a pretrained unconditional diffusion model to preliminarily purify the samples; (2) In the second stage, we leverage a stochastic refinement model to further adjust the samples, aligning them more closely with the original distribution. Inspired by recent progress in speech processing (Popov et al., 2021; Tian et al., 2023), we incorporate phoneme information into the refinement model to guide the refinement process.

Experimental results demonstrate that our method outperforms existing state-of-the-art adversarial purification methods in overcoming protective measures, enabling VC models to better capture authentic features. Our research suggests that current adversarial perturbation-based methods for VC defense may provide a false sense of security, inspiring the need for more robust solutions to address the security and privacy risks posed by advanced VC models.

Our contributions can be summarized as follows:

- We are the first to explore vulnerabilities of protective perturbation-based VC defenses, propose a realistic threat model, and systematically evaluate these defenses within it. We assess six VC methods and three protective techniques, revealing the risk that existing defenses potentially fail to prevent voice cloning attacks.

- To probe deeper into these risks, we propose a novel two-stage adversarial purification method to counter such protective techniques. Experimental results show that our method outperforms baselines in countering various protection methods across different cloning attacks, further exposing potential risks.

- We evaluate the robustness of our purification method against adaptive protections in a white-box setting. We

find that even with full access to the gradient information of our purification model, generating effective protective perturbations remains challenging, underscoring the need for more advanced techniques to prevent unauthorized data usage in VC.

## 2. Related Work

### 2.1. Voice Cloning

Voice cloning refers to the process of generating speech that imitates the voice of a specific target speaker. It can be achieved by text-to-speech (TTS) (Betker, 2023; Jia et al., 2018; Casanova et al., 2022) and voice conversion (Popov et al., 2022; Qin et al., 2023; Li et al., 2023a; Liu, 2024).

Both paradigms leverage the extracted speaker embeddings from the target speaker's audio to capture their vocal characteristics. In TTS, textual input serves as the content source. The TTS acoustic model, conditioned on the target speaker's embeddings, processes this text to generate corresponding acoustic features (*e.g.*, mel-spectrograms). These acoustic features are then used by a vocoder to synthesize speech in the target's voice. Alternatively, voice conversion typically utilizes acoustic features from a source speaker's utterance for the linguistic content. The voice conversion model then transforms these features using the target speaker's embeddings to generate speech in the target speaker's timbre while retaining the original linguistic content.

Recent advances in deep learning-based speech synthesis have enabled zero-shot synthesis—allowing the generation of speech for target speakers whose voices were not included in the training data. These zero-shot methods significantly increase risks by lowering the barrier to attacks, as attackers can easily access zero-shot VC services (*e.g.*, Elevenlabs (Prime Voice AI, 2025)) and open-source models without requiring specialized expertise. In response to these challenges, recent studies have explored countermesures against deep learning-based zero-shot VC (Huang et al., 2021; Li et al., 2023b; Yu et al., 2023). Building on these works, this study focuses on zero-shot VC models.

### 2.2. Protective Perturbations Against VC Attacks

To protect speech data from zero-shot VC attacks, recent studies have proposed adding imperceptible adversarial perturbations to disrupt the VC process. Huang et al. (2021) introduce the concept of adding such perturbations to speech to prevent VC models from synthesizing the voice of protected speaker. Li et al. (2023b) extended these perturbations to the time domain and incorporated a psychoacoustic model to ensure their imperceptibility. Yu et al. (2023) proposed an ensemble learning method to improve the transferability of these perturbations across different VC models, enabling them to generalize to unseen models. Additionally,

other studies (Liu et al., 2023; Yang et al., 2024; Wang et al., 2023; Dong et al., 2024) have explored similar methods to generate protective perturbations for speech data. While these methods have achieved notable success in preventing VC attacks, their effectiveness in real-world scenarios remains uncertain.

### 2.3. Adversarial Purification

Since protective perturbations can be viewed as a form of adversarial attack against VC models, a natural question arises: *can existing adversarial purification methods bypass these perturbations?* Adversarial purification aims to restore clean data by removing adversarial distortions. In the audio domain, adversarial purification methods can be broadly divided into two types: transformation-based approaches and reconstruction-based approaches. Transformation-based methods use techniques such as filtering, compression, and smoothing to disrupt adversarial perturbations (Hussain et al., 2021; Chen et al., 2023). These methods are easy to use, but not effective. On the other hand, reconstruction-based methods typically employ self-reconstruction models to restore clean audio. Recently, Wu et al. (2023) extended diffusion-based adversarial purification (Nie et al., 2022) to the audio domain, achieving state-of-the-art results in mitigating adversarial perturbations in speech command recognition tasks. Further research has explored diffusion-based adversarial purification using a combination of the time and frequency domains (Tan et al., 2024) and hierarchical purification (Guo et al., 2024). However, most of these adversarial purification methods are designed for classification tasks, including speaker recognition and speech recognition. Therefore, these methods tend to preserve only coarse feature information, leading to distortions in VC model embedding spaces, as illustrated in Figure 2.

## 3. Threat Model

**Overview of the Conflict.** Given a target speaker $T$ with original voice samples $x$, the attacker may use a voice cloning model $M$ to synthesize a forged voice $M(x)$ that mimics the voice of $T$. To counter this, the protector adds an imperceptible perturbation $\delta$ to $x$, creating a protected voice $x' = x + \delta$. The perturbation $\delta$ is bounded by a budget $\epsilon$, ensuring that $x'$ remains perceptually similar to $x$, but prevents attackers from using $x'$ to generate a convincing forgery of $T$. Therefore, the protector aims to achieve:

$$\text{H}(x') \approx \text{H}(x),$$
$$\text{H}(M(x')) \neq \text{H}(x) \quad \text{and} \quad \text{SV}(M(x')) \neq \text{SV}(x), \tag{1}$$

where $\text{H}(\cdot)$ represents the perceived speaker identity evaluated by human listeners with given audio, and $\text{SV}(\cdot)$ represents the evaluation by speaker verification (SV) systems.

The goal of the attacker is to bypass the protective pertur-

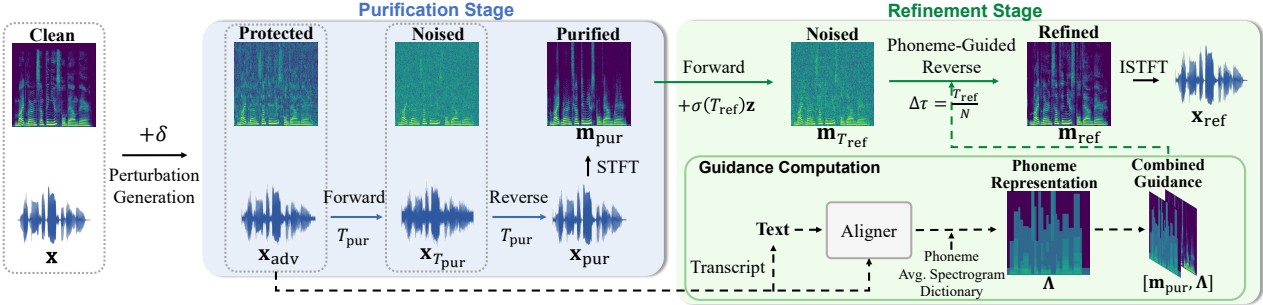

*Figure 3.* Inference process of our Purification-Refinement framework (gray dotted box: waveforms and corresponding spectrograms).

bations and generate a forged voice $M(p(x'))$, where $p(\cdot)$ denotes a potential purification function used to mitigate the perturbations in $x'$. The attacker aims to achieve:

$$\mathrm{H}(M(p(x'))) \approx \mathrm{H}(x) \ \text{ or } \ \mathrm{SV}(M(p(x'))) = \mathrm{SV}(x), \quad (2)$$

where the specific choice of targeting humans or SV systems depends on the intent of the attacker.

**Capabilities of the Attacker.** The attacker cannot directly record the original voice samples $x$ of the target speaker $T$, but can collect limited samples $x'$ from public-domain sources such as social media, video platforms, etc. Meanwhile, they can employ existing voice cloning models to perform zero-shot voice cloning with $x'$. The attacker **does not** have the knowledge of the protection methods applied by the protector, but they may observe perceptible artifacts in $x'$. Alternatively, unsatisfactory results from initial cloning attempts may also suggest the presence of protective measures. Therefore, they will preprocess $x'$ to mitigate protective perturbations before cloning.

**Capabilities of the Protector.** The protector may recognize that attackers attempt purification strategies to bypass the protection mechanisms. We consider two scenarios: (1) *Non-Adaptive Protection*, where the protector knows the structure and gradients of the voice cloning model used by the attacker, but does not consider the potential purification strategies. (2) *Adaptive Protection*, where the protector additionally knows the purification strategies of the attacker, including the structure and gradients of the purification model.

## 4. Proposed Method

In this section, we introduce our purification method, denoted as **PhonePuRe**. It primarily consists of two components: **Pu**rification and **Phone**me-Guided **Re**finement.

### 4.1. Overview: Purification-Refinement Framework

**Embedding Distortions in Existing Methods.** We first analyze existing purification methods. Since no prior work specifically addresses the purification of protective perturba-

tions for voice cloning tasks, we apply existing adversarial purification methods designed for speech classification tasks to this problem. The results are visualized in Figure 2.

We observed that while these methods reduce adversarial noise (evidenced by clustering samples of the same class), they also introduce systematic distortions in the VC model embedding space. Specifically, (1) samples from different classes become closer, and (2) purified samples deviate further from their original clean versions. Since VC models rely on embeddings to extract the speech characteristics of the target speaker, such distortions lead to an inability to capture accurate features of the speaker, which impacts the performance of VC.

The embedding distortions in current methods can be attributed to the properties of unconditional diffusion models used in purification. When diffusion steps are few, the model fails to fully purify samples. Conversely, with enough diffusion steps, the details of the samples are lost, causing the model to generate similar samples, as shown in Figure 2. In essence, the distribution of samples generated by the diffusion model differs from that of clean samples, leading to suboptimal performance in VC tasks.

**Our Proposed Purification-Refinement Framework.** Building on the insight above, we propose a two-stage framework consisting of a *Purification* stage and a *Phoneme-Guided Refinement* stage. The goal of the first stage is to preliminarily mitigate the adversarial noise, and the second stage employs a phoneme-guided diffusion model to further refine the purified samples, aligning them more closely with the clean distribution. Figure 3 illustrates the inference process of our proposed framework.

The Refinement stage is motivated by the challenge of directly mapping adversarial samples to clean samples due to the unknown distribution of adversarial perturbations. To address this, our approach is based on a key observation regarding our initial Purification stage. As illustrated in Figure 4, the Purification stage results in purified clean samples and purified protected samples exhibiting similar distributions. This similarity enables the Refinement model

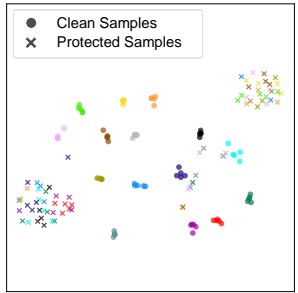 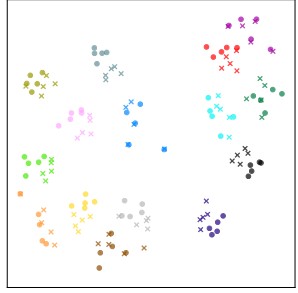

(a) Clean vs. Protected
(Before Purification)

(b) Purified (Clean) vs. Purified
(Protected)

*Figure 4.* Distribution comparison of clean and protected samples (a) **Before** and (b) **After** our initial Purification stage in the VC model embedding space. Different colors represent different speakers. The distribution of clean and protected samples becomes closer from (a) to (b), indicating that this type of diffusion-based purification leads to similar distributions for clean and protected samples, and provides an accessible starting point for the Refinement stage.

to be trained using only pairs of original clean samples and purified clean samples. Since purified clean data used for training is distributionally similar to the purified protected data encountered during inference, a mapping learned from the purified clean samples to the true clean distribution can then be applied to the purified protected samples. Such an application effectively maps these purified protected samples toward the clean distribution, aiming to reduce the embedding distortion common in existing methods.

### 4.2. Purification: Unconditional Diffusion

During the Purification stage, we employ the model proposed by Kong et al. (2021), which applies the diffusion process directly to audio waveforms. Given an input adversarial audio waveform $\mathbf{x}_{\mathrm{adv}}$, the Purification stage $P(\cdot)$ uses both the forward and reverse diffusion processes to produce the purified waveform $\mathbf{x}_{\mathrm{pur}}$. In the forward diffusion process, noise is progressively added to the initial waveform $\mathbf{x}_{\mathrm{adv}}$ over $T_{\mathrm{pur}}$ diffusion steps to form the final noised $\mathbf{x}_{T_{\mathrm{pur}}}$. At each timestep $t$, the process is expressed as:

$$q(\mathbf{x}_t|\mathbf{x}_{t-1}) = \mathcal{N}(\mathbf{x}_t; \sqrt{1-\beta_t}\mathbf{x}_{t-1}, \beta_t\mathbf{I}), \\ t = 1, 2, \ldots, T_{\mathrm{pur}}, \tag{3}$$

where $\beta_t$ denotes the variance schedule, and $\mathbf{x}_0 = \mathbf{x}_{\mathrm{adv}}$. The reverse process denoises the waveform $\mathbf{x}_{T_{\mathrm{pur}}}$ in the same $T_{\mathrm{pur}}$ steps, and at each step, it is formulated as:

$$\mathbf{x}_{t-1} \sim p_\theta(\mathbf{x}_{t-1}|\mathbf{x}_t) = \mathcal{N}(\mathbf{x}_{t-1}; \boldsymbol{\mu}_\theta(\mathbf{x}_t, t), \sigma_t^2\mathbf{I}), \\ t = T_{\mathrm{pur}}, T_{\mathrm{pur}} - 1, \ldots, 1, \tag{4}$$

where $\boldsymbol{\mu}_\theta(\mathbf{x}_t, t)$ is the mean function parameterized by $\theta$, $\sigma_t^2$ is the time-dependent variance schedule, and $\mathbf{x}_0 = \mathbf{x}_{\mathrm{pur}}$.

The entire Purification stage can be viewed as a sequence of diffusion and denoising steps, formalized as:

$$\mathbf{x}_{\mathrm{pur}} = P_\theta(\mathbf{x}_{\mathrm{adv}}), \tag{5}$$

where $\theta$ denotes the parameters of the Purification stage.

### 4.3. Refinement: Phoneme-Guided Score-Based Diffusion

The Refinement stage operates in the complex spectrogram domain. Given a set of clean speech samples $\{\mathbf{x}^{(i)}\}_{i=1}^N$, we obtain their purified versions $\mathbf{x}_{\mathrm{pur}}^{(i)} = P_\theta\left(\mathbf{x}^{(i)}\right)$. We then create a dataset $\mathcal{D} = \{(\mathbf{x}^{(i)}, \mathbf{x}_{\mathrm{pur}}^{(i)})\}_{i=1}^N$ by pairing clean and purified speech, which correspond to their complex spectrograms denoted as $\mathcal{D}_{\mathbb{C}} = \{(\mathbf{m}^{(i)}, \mathbf{m}_{\mathrm{pur}}^{(i)})\}_{i=1}^N$, where $\mathbf{m} = \mathrm{STFT}(\mathbf{x})$ (Short-Time Fourier Transform). The goal of the Refinement stage is to learn the conditional distribution $p_\phi(\mathbf{m}|\mathbf{m}_{\mathrm{pur}})$ over $\mathcal{D}_{\mathbb{C}}$. To achieve this, we employ a phoneme-guided score-based diffusion model to learn a parameterized approximation of $p_\phi(\mathbf{m}|\mathbf{m}_{\mathrm{pur}})$.

**Phoneme Representation.** The intuition behind using phonemes as guidance is based on the observation that protective perturbations against VC models are primarily designed to disrupt the speaker-specific characteristics of the audio, while phonemes encode the content information of speech. Since these protective perturbations are typically imperceptible and not explicitly designed to interfere with the speech content, they are likely to have a minimal impact on the phoneme information. Thus, phoneme information can be utilized as a clue to guide the Refinement phase.

Specifically, we represent phonemes using the average magnitude spectrogram, denoted as $\boldsymbol{\Lambda}$, which has the same shape as the linear magnitude spectrogram $|\mathbf{m}|$ of the audio $\mathbf{x}$. To compute $\boldsymbol{\Lambda}$, three steps are followed: (1) First, obtain the text transcription for each training sample. Then, perform phoneme alignment using the forced aligner (an acoustic model which aligns text to audio at the phoneme level, implemented in tools like Montreal Forced Aligner (MFA) (McAuliffe et al., 2017)) to convert the transcription of each training sample into a time-aligned phoneme sequence; (2) Next, compute the linear magnitude spectrograms of all training samples; (3) For each phoneme, calculate the average linear magnitude spectrogram across all instances of that phoneme in the training set as its representation. These representations are stored in an average phoneme dictionary.

In the Refinement stage, the audio samples and their corresponding text are set as input. Then we use forced aligner to perform phoneme alignment and look up the phoneme dictionary to retrieve the average spectrograms for the aligned phonemes. These averaged values are concatenated along

the time axis to form a phoneme representation $\mathbf{\Lambda}$.

**Score-Based Diffusion.** To incorporate phoneme information, we concatenate the phoneme representation $\mathbf{\Lambda}$ with the complex spectrogram $\mathbf{m}_{\text{pur}}$ to form the guiding input $[\mathbf{m}_{\text{pur}}, \mathbf{\Lambda}]$ for the Refinement stage. We employ a score-based diffusion model based on the Ornstein-Uhlenbeck Stochastic Differential Equation (OUSDE) following Welker et al. (2022). During training, given a clean sample $\mathbf{m}_0$ and its purified version $\mathbf{m}_{\text{pur}}$, the distribution of the diffused state $\mathbf{m}_\tau$ can be expressed as:

$$\mathbf{m}_\tau = \boldsymbol{\mu}(\mathbf{m}_0, [\mathbf{m}_{\text{pur}}, \mathbf{\Lambda}], \tau) + \sigma(\tau)\mathbf{z}, \qquad (6)$$

where $\boldsymbol{\mu}$ and $\sigma$ are defined following the variance-exploding scheme of Särkkä & Solin (2019), and $\mathbf{z} \sim \mathcal{N}_{\mathbb{C}}(0, \mathbf{I})$. Therefore, the denoising score matching objective is used to train the score model $s_\phi$ (Song et al., 2021):

$$\mathcal{L}(\phi) = \mathbb{E}\left[\left\|s_\phi(\mathbf{m}_\tau, [\mathbf{m}_{\text{pur}}, \mathbf{\Lambda}], \tau) + \frac{\mathbf{z}}{\sigma(\tau)}\right\|_2^2\right]. \quad (7)$$

where we employ a score estimator $s_\phi$ based on the NCSN++ architecture (Song et al., 2021), and $\phi$ denotes the parameters of the Refinement stage.

During inference, we perform forward diffusion on $\mathbf{m}_{\text{pur}}$ to obtain the noised state $\mathbf{m}_{T_{\text{ref}}} = \mathbf{m}_{\text{pur}} + \sigma(T_{\text{ref}})\mathbf{z}$, where $\mathbf{z} \sim \mathcal{N}_{\mathbb{C}}(0, \mathbf{I})$, and $T_{\text{ref}}$ represents the final time step for the Refinement process. We then perform reverse sampling with step size $\Delta\tau = \frac{T_{\text{ref}}}{N}$, where $N$ is the total number of reverse steps. At each discrete step, we use a predictor-corrector sampling scheme, a discrete sampling method, corrected by one-step annealed Langevin dynamics (Song et al., 2021), to compute $\mathbf{m}_{\tau-\Delta\tau}$ using the score function of $\mathbf{m}_\tau$, yielding the refined spectrogram $\mathbf{m}_{\text{ref}}$. Finally, $\mathbf{m}_{\text{ref}}$ is converted back to the time domain through the inverse STFT (ISTFT) to obtain the refined audio waveform $\mathbf{x}_{\text{ref}}$. The overall Refinement stage $R(\cdot)$ can be expressed as:

$$\mathbf{x}_{\text{ref}} = R_\phi(\mathbf{x}_{\text{pur}}, \mathbf{\Lambda}) = R_\phi(P_\theta(\mathbf{x}_{\text{adv}}), \mathbf{\Lambda}), \qquad (8)$$

where the refined audio waveform $\mathbf{x}_{\text{ref}}$ is the final output of our whole purification method.

## 5. Experiments

### 5.1. Experimental Setup

**Voice Cloning Methods.** We selected six advanced VC methods for evaluation, including three TTS models: YourTTS (Casanova et al., 2022), SV2TTS (Jia et al., 2018), and TorToise (Betker, 2023); and three voice conversion models: DiffVC (Popov et al., 2022), OpenVoice V2 (Qin et al., 2023), and SeedVC (Liu, 2024). Implementation details are provided in App. A.1.

**Protection Methods.** We evaluated three mainstream protective perturbation methods: AttackVC (Huang et al., 2021), AntiFake (Yu et al., 2023) and VoiceGuard (Li et al., 2023b), which achieve the highest protection success rates in our experiments. Details of the protection methods can be found in App. A.3.

**Adversarial Purification Baselines.** We compare our method against five recent adversarial purification methods: the transformation-based WaveGuard (Hussain et al., 2021) and SpeakerGuard (Chen et al., 2023), and the reconstruction-based AudioPure (Wu et al., 2023), WavePurifier (Guo et al., 2024), and DualPure (Tan et al., 2024). Details are provided in App. A.4.

**Evaluation Dataset.** The evaluation set consists of 25 speakers from the *test-clean* subset of LibriSpeech (Panayotov et al., 2015), each contributing 5 sentences. These speakers do not overlap with those in the training set of the adversarial purification models. We apply all six VC methods to the evaluation set, resulting in 750 ($25 \times 5 \times 6$) synthetic speech samples. Following previous work (Yu et al., 2023; Huang et al., 2021; Li et al., 2023b), we filter 739 synthetic speech samples that successfully pass at least one SV system for subsequent evaluation.

**Evaluation Metrics.** Our evaluation includes both objective and subjective metrics. Objective metrics include speaker verification accuracy (SVA) for effectiveness and objective mean opinion score (MOS) for naturalness. The SVA component is measured using x-vector-based SV (xSVA) (Desplanques et al., 2020) and d-vector-based SV (dSVA) (Wan et al., 2018), calculated as:

$$\text{SVA} = \frac{1}{N}\sum_{i=1}^{N}\text{SV}(\mathbf{x}_{\text{cloned}}^i), \qquad (9)$$

where $\mathbf{x}_{\text{cloned}}^i$ is the $i$-th cloned speech sample, $N$ is the total number of cloned samples, and $\text{SV}(\cdot)$ is the binary decision by x-vector or d-vector-based SV model. For the objective MOS, we employ NISQA (Mittag et al., 2021), a neural network-based model for objective audio quality assessment, which evaluates overall quality and naturalness on a scale of 1 to 5, with higher scores indicating better quality. Subjective metrics involve perceived speaker similarity, assessed by human listeners who determine if two samples are from the same speaker using a four-level scale: Same (Certain), Same (Uncertain), Different (Uncertain), and Different (Certain). Details are provided in App. A.6.

**Training Details.** Our method trains two models separately, and cascades them during inference. The Purification model is based on a pretrained unconditional DiffWave model (Kong et al., 2021) which is then fine-tuned on the LibriSpeech (Panayotov et al., 2015) dataset in the time

*Table 1.* Speaker verification accuracy of synthesized speech on the evaluation dataset.

| Protection Method | VC Method | Protected | | WaveGuard | | SpeakerGuard | | DualPure | | WavePurifier | | AudioPure | | **Ours** | |
|---|---|---|---|---|---|---|---|---|---|---|---|---|---|---|---|
| | | xSVA | dSVA | xSVA | dSVA | xSVA | dSVA | xSVA | dSVA | xSVA | dSVA | xSVA | dSVA | xSVA | dSVA |
| AntiFake | YourTTS | 0.034 | 0.029 | 0.328 | 0.155 | 0.277 | 0.136 | 0.067 | 0.025 | 0.286 | 0.301 | 0.597 | 0.417 | **0.672** | **0.689** |
| | SV2TTS | 0.019 | 0.000 | 0.038 | 0.008 | 0.038 | 0.033 | 0.067 | 0.000 | 0.298 | 0.264 | 0.279 | 0.306 | **0.654** | **0.744** |
| | Tortoise | 0.059 | 0.042 | 0.051 | 0.025 | 0.034 | 0.025 | 0.025 | 0.000 | 0.203 | 0.167 | 0.441 | 0.475 | **0.627** | **0.725** |
| | DiffVC | 0.029 | 0.040 | 0.048 | 0.032 | 0.096 | 0.113 | 0.019 | 0.016 | 0.317 | 0.379 | 0.231 | 0.331 | **0.673** | **0.823** |
| | OpenVoice* | 0.433 | 0.405 | 0.327 | 0.207 | 0.240 | 0.241 | 0.048 | 0.026 | 0.337 | 0.190 | 0.471 | 0.534 | **0.625** | **0.776** |
| | SeedVC* | 0.331 | 0.455 | 0.153 | 0.211 | 0.395 | 0.325 | 0.000 | 0.000 | 0.355 | 0.447 | 0.363 | 0.642 | **0.702** | **0.805** |
| | Avg. | 0.152 | 0.164 | 0.159 | 0.105 | 0.186 | 0.146 | 0.037 | 0.011 | 0.299 | 0.293 | 0.401 | 0.451 | **0.660** | **0.762** |
| AttackVC | YourTTS | 0.108 | 0.000 | 0.571 | 0.359 | 0.605 | 0.544 | 0.412 | 0.350 | 0.588 | 0.612 | 0.739 | 0.515 | **0.748** | **0.709** |
| | SV2TTS | 0.127 | 0.140 | 0.173 | 0.107 | 0.221 | 0.273 | 0.067 | 0.074 | 0.510 | 0.636 | 0.731 | 0.868 | **0.740** | **0.893** |
| | Tortoise | 0.131 | 0.092 | 0.195 | 0.175 | 0.381 | 0.350 | 0.034 | 0.000 | 0.508 | 0.283 | 0.746 | 0.833 | **0.754** | **0.883** |
| | DiffVC | 0.173 | 0.282 | 0.231 | 0.169 | 0.260 | 0.476 | 0.000 | 0.000 | 0.587 | 0.661 | 0.817 | 0.903 | **0.846** | **0.935** |
| | OpenVoice | 0.000 | 0.000 | 0.394 | 0.293 | 0.327 | 0.345 | 0.048 | 0.043 | 0.481 | 0.336 | 0.635 | 0.724 | **0.663** | **0.862** |
| | Avg. | 0.108 | 0.108 | 0.317 | 0.216 | 0.366 | 0.394 | 0.118 | 0.086 | 0.536 | 0.505 | 0.734 | 0.777 | **0.750** | **0.861** |
| VoiceGuard | YourTTS | 0.050 | 0.000 | 0.521 | 0.136 | 0.462 | 0.330 | 0.378 | 0.282 | 0.496 | 0.563 | 0.672 | 0.417 | **0.697** | **0.621** |
| | SV2TTS | 0.058 | 0.058 | 0.106 | 0.033 | 0.115 | 0.017 | 0.067 | 0.282 | 0.375 | 0.421 | 0.596 | 0.752 | **0.731** | **0.851** |
| | Tortoise | 0.042 | 0.008 | 0.161 | 0.175 | 0.136 | 0.083 | 0.025 | 0.000 | 0.407 | 0.242 | 0.712 | 0.742 | **0.754** | **0.892** |
| | DiffVC | 0.029 | 0.121 | 0.173 | 0.121 | 0.135 | 0.185 | 0.000 | 0.000 | 0.510 | 0.508 | 0.683 | 0.847 | **0.750** | **0.903** |
| | OpenVoice | 0.000 | 0.000 | 0.327 | 0.164 | 0.212 | 0.224 | 0.067 | 0.034 | 0.317 | 0.207 | 0.606 | 0.759 | **0.683** | **0.853** |
| | Avg. | 0.036 | 0.039 | 0.262 | 0.125 | 0.217 | 0.163 | 0.113 | 0.115 | 0.423 | 0.385 | 0.656 | 0.712 | **0.723** | **0.830** |
| Total | | 0.099 | 0.104 | 0.246 | 0.148 | 0.256 | 0.234 | 0.089 | 0.070 | 0.419 | 0.394 | 0.597 | 0.647 | **0.711** | **0.818** |

The asterisk (*) indicates black-box senario (see App. A.3). Best performance is highlighted in **bold**.

domain. The Refinement model is trained on pairs of the original clean samples and purified samples derived from an augmented LibriSpeech dataset as described in section 4.3. The Refinement stage operates in the complex spectrogram domain, utilizing a 16 kHz STFT with a window size of 510, hop length of 128, and square-root Hann window. Details of the training process can be found in App. A.5.

### 5.2. Main Results

**Objective Evaluation.**

*Existing Protection and Purification Methods.* We first evaluate existing methods from the perspective of effectiveness under the threat model in section 3. Table 1 shows the xSVA and dSVA of speech synthesized using protected or purified samples in two SV systems. We present the results of the best-performing purification methods from each paper; other results are provided in App. B.1.

We find that without adversarial purification, all protection methods successfully reduce the SVA to below 20% in a white-box setting. However, when using purification methods, we observe a significant increase in the SVA. For instance, AudioPure can increase the xSVA to an average of 59.7%. This indicates that existing adversarial purification methods can neutralize protective perturbations, allowing attackers to successfully clone the protected speech.

Nevertheless, we find that existing methods still result in a relatively low SVA under certain protection methods. For example, AudioPure has a dSVA of only 45.1% against the

AntiFake protection. Additionally, when listening to these purified samples, they appear muffled and contain noticeable artifacts, indicating that existing methods introduce distortions, which likely contribute to the lower SVA.

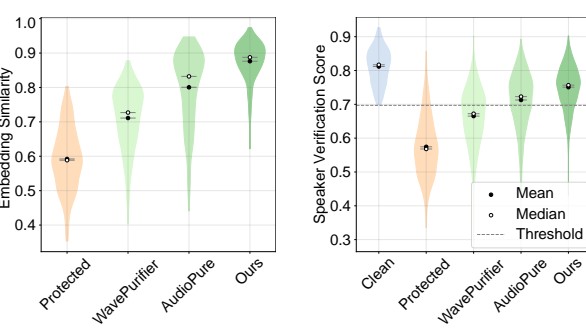

(a) VC embedding similarity  (b) Synthesized speech SV score

*Figure 5.* Distribution of (a) Cosine similarity between clean and protected/purified samples in VC model embedding space; (b) Speaker verification scores for synthesized speech using clean, protected, and purified samples.

*Our Purification Method.* We first examine whether our method reduces the distortion in VC model embedding space. As illustrated in Figure 5a and Figure 2d, embeddings of purified samples are brought closer to those of clean samples, demonstrating effective distortion reduction.

We further evaluate the effectiveness of our method by analyzing speaker verification performance. As shown in Table 1, our method achieves the highest xSVA and dSVA across all VC models and protection methods. In particular,

*Table 2.* Ablation study of our components in different protection methods.

| Method | Purification Stage | Refinement Stage | Phoneme Guidance | AntiFake xSVA | AntiFake dSVA | AttackVC xSVA | AttackVC dSVA | VoiceGuard xSVA | VoiceGuard dSVA | **Total** xSVA | **Total** dSVA |
|---|---|---|---|---|---|---|---|---|---|---|---|
| w/o Purification | ✗ | ✓ | ✓ | 0.324 | 0.339 | 0.446 | 0.500 | 0.279 | 0.274 | 0.350 | 0.371 |
| w/o Refinement | ✓ | ✗ | ✗ | 0.401 | 0.451 | 0.734 | 0.776 | 0.656 | 0.712 | 0.597 | 0.646 |
| w/o Phoneme | ✓ | ✓ | ✗ | 0.632 | 0.719 | 0.743 | 0.829 | 0.710 | 0.810 | 0.695 | 0.786 |
| Full model | ✓ | ✓ | ✓ | **0.660** | **0.762** | **0.750** | **0.861** | **0.723** | **0.830** | **0.711** | **0.818** |

*Table 3.* Objective MOS of synthesized speech.

| Clean | Protected | AudioPure | WavePurifier | **Ours** |
|---|---|---|---|---|
| $3.42 \pm 0.59$ | $3.16 \pm 0.65$ | $3.14 \pm 0.55$ | $3.34 \pm 0.67$ | **$3.36 \pm 0.58$** |

our method achieves a dSVA of 76.2% on the AntiFake protection method, surpassing existing methods by at least 31.1%. Figure 5b compares the distribution of speaker verification scores for speech synthesized from various samples. Compared to existing methods, our method's distribution is closer to that of clean speech, which aligns with the improved fidelity observed in VC model embedding space, confirming that our method significantly enhances the ability of VC models to replicate the target speaker identity.

*Quality of Synthesized Speech.* Table 3 shows the objective MOS scores for speech synthesized from samples purified using our method and existing purification methods. The results demonstrate that our method yields greater naturalness, enhancing the ability of the VC model to both replicate the target speaker identity and produce more natural speech.

**Subjective Evaluation.**

We follow previous work (Huang et al., 2021) and conduct a listening test with 20 participants, who are asked to rate the perceived speaker similarity between the original clean speech and speech synthesized using clean, protected, and purified samples. For each pair of utterances, participants decide if the two utterances are from the same speaker by selecting one of four options: Same (Certain), Same (Uncertain), Different (Uncertain), and Different (Certain). Each participant is asked to assess 40 speech pairs, resulting in a total of 800 evaluations.

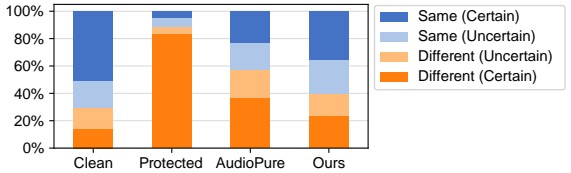

*Figure 6.* Perceived speaker similarity between original clean speech and synthesized speech using clean, protected, and purified samples, as assessed by human listeners. Audio samples are available online: https://de-antifake.github.io/samples.

Figure 6 presents our subjective evaluation results. The results indicate that the speech synthesized from purified voices not only bypasses SV systems but also exhibits higher speaker similarity in subjective assessments, demonstrating potential risks of VC attacks. Furthermore, our purification method outperforms existing methods in enhancing the cloned speech's perceptual similarity to the original speaker, thereby increasing the potential for such attacks.

### 5.3. Ablation Study

**Purification-Refinement Framework.** We first evaluated the effectiveness of our two-stage framework, as shown in Table 2. Compared to the full framework, removing either the Purification or Refinement stage resulted in worse performance. Without the Refinement stage, the performance decreased, emphasizing its contribution to improvement. Conversely, omitting the Purification stage led to much lower performance, indicating that the Refinement model fails to neutralize perturbations without prior Purification stage. This suggests that the Purification stage neutralizes perturbations, while the Refinement stage enhances performance, confirming the effectiveness of our framework.

**Phoneme-Guidance in Refinement.** We further evaluated the impact of phoneme guidance in the Refinement phase on our method, as shown in Table 2. We found that introducing phoneme guidance improved SVA. Moreover, even without phoneme guidance, our method outperforms existing methods under all protection strategies.

**Perturbation Budget and Purification Steps.** Since our Purification and Refinement stages are cascaded, a natural question is *whether our method simply improves performance by increasing the number of diffusion steps*. Figure 7 shows the performance of our method against the AttackVC protection method under different perturbation budgets $\epsilon$ and Purification stage diffusion steps $T_{\text{pur}}$.

We find that, for the same perturbation budget, as the number of Purification stage diffusion steps $T_{\text{pur}}$ increases, the purification performance initially improves and then decreases. This aligns with our intuition, as more diffusion steps dilute both the perturbations and the sample details, making purified samples blurry or overly smooth. For smaller perturbation budgets, fewer diffusion steps generally perform better, while larger budgets favor more steps.

Moreover, our full two-stage model outperforms the Purification stage alone across all perturbation budgets and diffusion steps, even when the perturbation budget is small. This suggests that performance improvement comes not from adding more diffusion steps (since Purification stage does not achieve the same performance gains with an increased number of steps), but from aligning the purified distribution with the clean distribution via Refinement, further confirming the effectiveness of our framework.

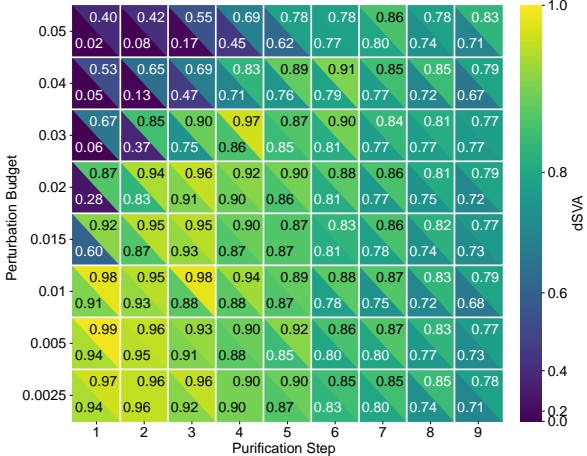

*Figure 7.* Impact of perturbation budget and diffusion steps of Purification stage on our method. In each grid, the lower-left corner represents the Purification stage only, while the upper-right corner shows the results for the full two-stage framework.

### 5.4. Adaptive Protection

**Adaptive Protection Strategies.** Adaptive protection enhances the robustness of perturbations by considering the impact of purification methods. In a white-box scenario, where the protector has full access to the attacker's VC model, purification methods, and gradients, a common approach is integrating the purification into the perturbation generation pipeline to optimize with gradients.

However, the iterative nature of the diffusion models in our Purification and Refinement stages creates deep computational graphs, leading to high memory costs, vanishing gradients, and exploding gradients (Lee & Kim, 2023a; Kang et al., 2024), complicating gradient computation. To overcome these challenges, we use two approximation methods:

• *Backward Pass Differentiable Approximation (BPDA) Adaptive Protection.* We integrate the full purification method into the perturbation generation pipeline. Following Nie et al. (2022), we treat the purification process as an identity transformation during backpropagation, allowing us to compute approximate gradients and optimize perturbations end-to-end. Due to the randomness in the diffusion and reverse processes, we apply Expectation Over Transformation

(EOT) on BPDA adaptive protection, using EOT sizes of 1, 5, 10, and 15. We use the AttackVC protection method and implement 150-step BPDA+EOT adaptive protection for the DiffVC model.

• *Adjoint Gradient-Based Adaptive Protection.* We integrate the Purification stage into the perturbation generation pipeline, using the adjoint method from Wu et al. (2023) to calculate the gradients of the Purification stage, avoiding out-of-memory issues. Other experimental settings remain consistent with the BPDA adaptive protection.

**Adaptive Protection Results.** We implement two adaptive protection strategies for different purification methods respectively, purify the generated adaptive protection samples, and then use the purified samples to perform the VC process. The resulting dSVA of the synthesized speech is shown in Figure 8. We observe that under both adaptive protection strategies, our method demonstrates greater robustness than existing methods. The small variation in dSVA across different EOT sizes indicates that EOT plays a limited role in generating effective adaptive perturbations. Even at an EOT size of 15, our method maintains a dSVA above 0.8, indicating that a substantial proportion of synthesized speech using the purified samples can still evade speaker verification. Consequently, even under white-box conditions, designing adaptive protection for our purification method remains a challenge for protectors, highlighting the risks we have identified.

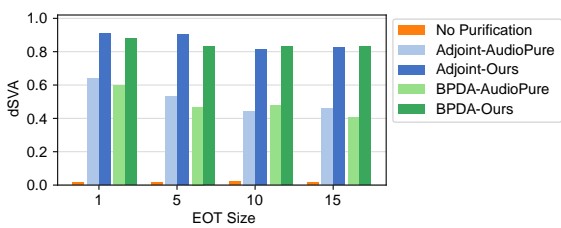

*Figure 8.* Robust dSVA under different adaptive strategies.

## 6. Conclusion

In this paper, we conduct the first systematic evaluation of protective perturbation-based VC defenses under realistic threat models that include perturbation purification. Our experiments demonstrate that attackers can use purification methods to bypass most of these protections and successfully synthesize speech that can deceive both speaker verification systems and human perception. Furthermore, we introduce a novel two-stage adversarial purification method, proposing a Purification-Refinement framework that incorporates phoneme guidance during the Refinement stage. Experimental results demonstrate that our method outperforms state-of-the-art methods in bypassing perturbation-based VC defenses and is robust against adaptive protections.

## Acknowledgements

This work was supported in part by the National Natural Science Foundation of China under Grant 62472398, U2336206, U2436601, and 62121002.

## Impact Statement

This study aims to assess the limitations of existing voice cloning defenses based on protective perturbations and proposes a novel adversarial purification method to undermine these defenses. The societal implications of this work are complex and may raise potential ethical and social concerns. On one hand, our findings contribute to a deeper understanding of the vulnerabilities in current VC defenses, which can inform the development of more robust solutions to prevent malicious use of voice cloning technology. On the other hand, the technique we propose could be misused to bypass VC defenses, enabling malicious actors to clone voices more effectively for fraudulent or harmful purposes. This underscores the urgent need for ethical considerations, regulatory frameworks, and public awareness to mitigate the risks associated with voice cloning technology. We encourage collaboration between the research community and policymakers to address these challenges and ensure that advancements in this field are guided by principles of security, privacy, and societal well-being.

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

# A. Implement Details

## A.1. Implement Details of Voice Cloning

We selected six advanced VC methods for evaluation, including three TTS models: YourTTS (Casanova et al., 2022), SV2TTS (Jia et al., 2018), and TorToise (Betker, 2023); and three voice conversion models: DiffVC (Popov et al., 2022), OpenVoice V2 (Qin et al., 2023), and SeedVC (Liu, 2024).

We use official implementations and default parameters for all VC methods. For TTS models, we follow previous work (Yu et al., 2023) and use 24 sentences designed to simulate real-world threat scenarios as the synthetic speech content. For voice conversion models, we select random utterances from speakers in the evaluation set who are different from the current speaker as the synthetic speech content.

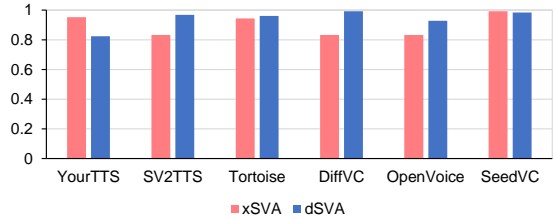

*Figure 9.* Effectiveness of VC models in bypassing the SV systems.

## A.2. Composition of the Evaluation Dataset

The evaluation set consists of 25 speakers from the *test-clean* subset of LibriSpeech (Panayotov et al., 2015), each contributing 5 sentences, ranging from short (2-4 seconds) to long (10-15 seconds). These speakers do not overlap with those in the training set of the purification models. We apply each of the six VC methods to the evaluation set, generating a total of 750 ($25 \times 5 \times 6$) synthetic speech samples. Figure 9 shows the xSVA and dSVA of synthesized speech from original clean samples using different VC models in our experiment, demonstrating the ability of each model to clone the target speaker's voice and their effectiveness in bypassing the SV system. In accordance with prior studies (Yu et al., 2023; Huang et al., 2021; Li et al., 2023b), we filter 739 synthetic speech samples that successfully pass at least one SV system for subsequent evaluation.

## A.3. Implement Details of Protection Methods

For methods with official implementations, we use their official code. For others, we reproduce them base on the original descriptions and adjust the parameters as needed to ensure an SVA of under 20% in a white-box setting.

- **AttackVC** (Huang et al., 2021): We implemented the default embedding defense (*emb*) method from the paper to defend against all VC models in a white-box

setting. For each VC model, we adjust the perturbation budget $\epsilon$ to ensure an SVA under 20%. However, for the SeedVC model, the embedding defense was unsuccessful, as this model not only relies on the speaker encoder for speaker information but also uses additional spectral features. Therefore, we only report the results for the other five VC models.

- **VoiceGuard** (Li et al., 2023b): We defend against all VC models in a white-box setting. For each VC model, we adjust the adaptive parameter $\alpha$, which balances stealth and protection capability, to ensure an SVA under 20%. For the same reasons as with AttackVC, the defense against the SeedVC model was unsuccessful. Therefore, we only present the results for the other five VC models.

- **AntiFake** (Yu et al., 2023): We used their default parameters and the combination of all four speaker encoders in the ensemble learning strategy, applying the target-based method from the paper to defend against all VC models. Since the encoders of the OpenVoice and SeedVC models are not included in their ensemble learning strategy, we present the results for these two models as **black-box** setting results in Table 1 and Table 5, while the results for the other four models are presented as white-box setting results.

## A.4. Implement Detials of Purification Baselines

For existing adversarial purification methods, we use their official implementations and default parameters for the experiments. For the reconstruction-based purification methods, we fine-tuned their official checkpoints on the LibriSpeech dataset for fair comparison.

We have listed the parameters of the existing purification methods used in our experiments in Table 4. For papers with multiple methods, the boldface represents the method with the highest average SVA in our experiments. We discussed the effectiveness of these methods in section 5.2, and other results are presented in App. B.1.

## A.5. Implement Details of Our Proposed Method

### Implement Details of the Purification Model.

The Purification model is based on a pretrained unconditional DiffWave model[1], which is then fine-tuned on the LibriSpeech (Panayotov et al., 2015) dataset for 16k steps with a learning rate of $10^{-4}$. Due to the input length limitation of the DiffWave model (16,000 samples), we trimmed all speech samples to a length of 16,000 and fed them individually into the Purification model. During the inference stage, the outputs were then concatenated to match the orig-

---

[1]https://github.com/philsyn/diffwave-unconditional

*Table 4.* Parameters of existing purification methods in our experiments.

| Baseline | Method Description | Parameters | Parameter Value |
|---|---|---|---|
| WaveGuard (Hussain et al., 2021) | **Mel Extraction - Inversion (Mel.)** | Number of mel bins | 80 |
| | Downsampling - Upsampling (DS) | Down-sampling rate (Hz) | 8000 |
| | Linear Predictive Coding (LPC) | LPC order | 20 |
| | Frequency Filtering (FF) | Negative gain magnitude (dB) | 30 |
| | Quantization - Dequantization (QT) | Number of quantization bits | 8 |
| SpeakerGuard (Chen et al., 2023) | **MPEG Compression (MP3-C)** | Bit rate (constant) | 16000 |
| | Average Smoothing (AS) | Kernel size | 3 |
| | Median Smoothing (MS) | Window size | 3 |
| | Low-Pass Filter (LPF) | Cut-off frequency (Hz) | 4000 |
| | | Stop-band frequency (Hz) | 8000 |
| | | Pass-band ripple (dB) | 3 |
| | | Stop-band attenuation (dB) | 40 |
| | Band-Pass Filter (BPF) | Pass-band edge frequencies (Hz) | [300, 4000] |
| | | Stop-band frequencies (Hz) | [50, 8000] |
| | | Pass-band ripple (dB) | 3 |
| | | Stop-band attenuation (dB) | 40 |
| AudioPure (Wu et al., 2023) | **DiffWave** | Number of reverse steps | 3 |
| | DiffSpec | Number of reverse steps | 3 |
| **WavePurifier** (Guo et al., 2024) | | Number of reverse steps on [0, 2000] Hz | 56 |
| | | Number of reverse steps on [2000, 4000] Hz | 40 |
| | | Number of reverse steps on [4000, 8000] Hz | 40 |
| **DualPure** (Tan et al., 2024) | | Number of reverse steps | 3 |

inal length. The Purification model operates at a sample rate of 16 kHz. For reverse diffusion, we set the number of Purification steps as $T_{\text{pur}} = 3$, and employ the Denoising Diffusion Probabilistic Models (DDPM) sampling method to generate the purified audio.

**Implementation Details of the Refinement Model.**

*Model Architecture and Diffusion Details.* For the Refinement model described in section 4.3, we utilize a score estimator based on the NCSN++ architecture (Song et al., 2021). The stiffness parameter is fixed at $\gamma = 1.5$, the extremal noise levels are set to $\sigma_{\min} = 0.05$ and $\sigma_{\max} = 0.5$, and the extremal diffusion times are set to $T = 1$ and $\tau_\epsilon = 0.03$. For reverse diffusion, we use $N = 30$ time steps and adopt the predictor-corrector scheme (Song et al., 2021), applying one step of annealed Langevin dynamics correction with a step size of $r = 0.4$.

*Data Representation.* The Refinement stage operates in the complex spectrogram domain, using STFT parameters with a window size of 510, a hop length of 128, and a square-root Hann window, all at a sample rate of 16 kHz. To compress the dynamic range of the input spectrograms, we apply square-root magnitude warping. During training, sequences of 256 STFT frames (approximately 2 seconds) are randomly sampled from full-length audio samples, normalized by the maximum absolute value of the purified samples, and then fed into the network.

*Data Augmentation.* For the training set of the Refinement model, we first add noise randomly selected from the DE-MAND dataset (Thiemann et al., 2013) to the *train-clean-100* subset of LibriSpeech (Panayotov et al., 2015), using only the first channel of the noise data. The Signal-to-Noise Ratio (SNR) levels for the added noise are randomly selected from $[0, 5, 10, 15, \text{None}]$, where None indicates that no noise is added to the sample. Then, we use the Purification model fine-tuned on the LibriSpeech dataset to purify these noisy samples and generate the purified data, where the Purification steps is set as $T_{\text{pur}} = 5$. The purified data is paired with the original clean data to form the training data pairs for the Refinement model. Note that the noise addition here is not intended for the Refinement model to learn denoising but serves as a data augmentation strategy designed to expose the Refinement model to various variants of the Purification distribution during training.

## A.6. Implement Details of Evaluation Metrics

**Speaker Verification Systems.** We employ two widely adopted SV systems: the x-vector-based SV[2] (xSVA) and the d-vector-based SV[3] (dSVA), which achieve equal error rates (EERs) of $0.018$ and $0.016$, respectively, on our evaluation dataset, representing their ability to distinguish between different speakers. For xSVA, we use the default threshold $0.25$. For dSVA, we set the threshold to $0.697$ using the EER criterion, as there is no default.

**Objective Mean Opinion Score.** We employ NISQA (Mittag et al., 2021), a neural network-based model for objective

---

[2]https://github.com/speechbrain/speechbrain
[3]https://github.com/resemble-ai/Resemblyzer

Table 5. Speaker verification accuracy of synthesized speech using more existing purification methods.

| Protection Method | VC Method | Purification Method | | | | | | | | | | | | | | | | |
|---|---|---|---|---|---|---|---|---|---|---|---|---|---|---|---|---|---|---|
| | | WaveGuard | | | | | | | | SpeakerGuard | | | | | | | | AudioPure | |
| | | QT | | LPC | | FF | | DS | | AS | | MS | | LPF | | BPF | | DiffSpec | |
| | | xSVA | dSVA | xSVA | dSVA | xSVA | dSVA | xSVA | dSVA | xSVA | dSVA | xSVA | dSVA | xSVA | dSVA | xSVA | dSVA | xSVA | dSVA |
| AntiFake | YourTTS | 0.042 | 0.049 | 0.118 | 0.068 | 0.050 | 0.029 | 0.151 | 0.078 | 0.034 | 0.010 | 0.168 | 0.107 | 0.034 | 0.019 | 0.059 | 0.010 | 0.048 | 0.050 |
| | SV2TTS | 0.019 | 0.017 | 0.077 | 0.000 | 0.029 | 0.025 | 0.058 | 0.017 | 0.019 | 0.025 | 0.038 | 0.008 | 0.038 | 0.008 | 0.048 | 0.025 | 0.067 | 0.000 |
| | Tortoise | 0.085 | 0.042 | 0.042 | 0.008 | 0.059 | 0.033 | 0.068 | 0.008 | 0.085 | 0.042 | 0.034 | 0.017 | 0.068 | 0.033 | 0.085 | 0.017 | 0.034 | 0.000 |
| | DiffVC | 0.048 | 0.040 | 0.058 | 0.000 | 0.202 | 0.048 | 0.048 | 0.024 | 0.096 | 0.065 | 0.077 | 0.048 | 0.019 | 0.040 | 0.096 | 0.048 | 0.000 | 0.008 |
| | OpenVoice* | 0.442 | 0.388 | 0.033 | 0.106 | 0.390 | 0.385 | 0.240 | 0.259 | 0.308 | 0.422 | 0.471 | 0.474 | 0.452 | 0.483 | 0.221 | 0.078 | 0.038 | 0.043 |
| | SeedVC* | 0.323 | 0.463 | 0.033 | 0.105 | 0.452 | 0.259 | 0.266 | 0.309 | 0.379 | 0.447 | 0.331 | 0.431 | 0.347 | 0.528 | 0.282 | 0.138 | 0.016 | 0.008 |
| | Avg. | 0.160 | 0.168 | 0.060 | 0.047 | 0.198 | 0.131 | 0.141 | 0.116 | 0.156 | 0.171 | 0.187 | 0.181 | 0.160 | 0.188 | 0.134 | 0.054 | 0.034 | 0.017 |
| AttackVC | YourTTS | 0.084 | 0.010 | 0.328 | 0.049 | 0.059 | 0.010 | 0.261 | 0.155 | 0.067 | 0.000 | 0.378 | 0.262 | 0.059 | 0.000 | 0.084 | 0.039 | 0.429 | 0.165 |
| | SV2TTS | 0.096 | 0.174 | 0.087 | 0.000 | 0.413 | 0.215 | 0.173 | 0.174 | 0.163 | 0.273 | 0.154 | 0.182 | 0.192 | 0.182 | 0.183 | 0.066 | 0.038 | 0.050 |
| | Tortoise | 0.237 | 0.275 | 0.085 | 0.008 | 0.381 | 0.150 | 0.339 | 0.200 | 0.322 | 0.150 | 0.297 | 0.375 | 0.254 | 0.208 | 0.263 | 0.075 | 0.059 | 0.008 |
| | DiffVC | 0.183 | 0.282 | 0.144 | 0.000 | 0.663 | 0.476 | 0.260 | 0.427 | 0.356 | 0.492 | 0.327 | 0.444 | 0.240 | 0.371 | 0.212 | 0.153 | 0.067 | 0.113 |
| | OpenVoice | 0.010 | 0.009 | 0.173 | 0.026 | 0.029 | 0.009 | 0.154 | 0.164 | 0.202 | 0.284 | 0.375 | 0.397 | 0.135 | 0.216 | 0.115 | 0.052 | 0.106 | 0.060 |
| | Avg. | 0.124 | 0.156 | 0.166 | 0.015 | 0.304 | 0.180 | 0.240 | 0.228 | 0.220 | 0.248 | 0.308 | 0.334 | 0.175 | 0.202 | 0.171 | 0.079 | 0.146 | 0.077 |
| VoiceGuard | YourTTS | 0.042 | 0.019 | 0.269 | 0.078 | 0.042 | 0.029 | 0.143 | 0.029 | 0.042 | 0.010 | 0.269 | 0.058 | 0.042 | 0.010 | 0.067 | 0.029 | 0.202 | 0.039 |
| | SV2TTS | 0.096 | 0.025 | 0.048 | 0.000 | 0.173 | 0.066 | 0.077 | 0.033 | 0.135 | 0.058 | 0.077 | 0.033 | 0.067 | 0.050 | 0.058 | 0.033 | 0.077 | 0.058 |
| | Tortoise | 0.085 | 0.042 | 0.119 | 0.008 | 0.229 | 0.050 | 0.161 | 0.042 | 0.186 | 0.033 | 0.136 | 0.075 | 0.195 | 0.050 | 0.144 | 0.050 | 0.042 | 0.000 |
| | DiffVC | 0.058 | 0.145 | 0.125 | 0.000 | 0.346 | 0.274 | 0.144 | 0.113 | 0.192 | 0.177 | 0.135 | 0.218 | 0.067 | 0.105 | 0.221 | 0.089 | 0.048 | 0.016 |
| | OpenVoice | 0.000 | 0.000 | 0.106 | 0.009 | 0.029 | 0.000 | 0.087 | 0.138 | 0.135 | 0.198 | 0.221 | 0.216 | 0.096 | 0.060 | 0.048 | 0.009 | 0.067 | 0.060 |
| | Avg. | 0.056 | 0.048 | 0.137 | 0.017 | 0.162 | 0.087 | 0.124 | 0.072 | 0.137 | 0.098 | 0.169 | 0.122 | 0.095 | 0.057 | 0.107 | 0.043 | 0.089 | 0.034 |
| Total | | 0.114 | 0.124 | 0.121 | 0.026 | 0.222 | 0.133 | 0.168 | 0.139 | 0.171 | 0.172 | 0.221 | 0.212 | 0.143 | 0.149 | 0.137 | 0.058 | 0.090 | 0.043 |

The asterisk (*) indicates black-box senario (see App. A.3).

audio quality assessment, which evaluates overall quality and naturalness on a 1-5 scale, with higher scores indicating better quality. For reference, the average objective MOS for the TIMIT dataset is $3.45 \pm 0.52$ (Yu et al., 2023). Our method achieves an average objective MOS of $3.36 \pm 0.58$ on the evaluation dataset as shown in Table 3, indicating that the synthesized speech is of high quality and naturalness.

**Subjective Evaluation.** We follow previous work (Huang et al., 2021) and conduct a listening test with 20 participants, who are asked to rate the perceived speaker similarity between the original clean speech and speech synthesized using clean, protected, and purified samples. For each pair of utterances, participants decide if the two utterances are from the same speaker by selecting one of four options: Same (Certain), Same (Uncertain), Different (Uncertain), and Different (Certain). Each participant is asked to assess 40 speech pairs, resulting in a total of 800 evaluations. The clean speech samples are randomly selected from our evaluation set. For each selected sample, a protection method is randomly assigned to generate the protected and purified versions. Subsequently, a randomly selected VC method is employed to generate the corresponding synthesized speech from clean, protected and purified versions. The subjective evaluation results are presented in Figure 6.

# B. Additional Experimental Results

## B.1. Additional Results on the Effectiveness of Existing Adversarial Purification Methods

We have listed the parameters of the existing purification methods in our experiments in Table 4. For papers with mul-

tiple methods, the boldface represents the method with the highest average SVA in our experiments. We have discussed the effectiveness of these methods in section 5.2. In Table 5, we provide the SVA results for other methods from these papers. We found that the average SVA of these methods ranges from 0.043 to 0.222, showing limited effectiveness in bypassing the protected perturbations. As a result, we did not discuss these methods in section 5.2.

The underperformance of transformation-based methods may be due to simple transformations failing to distinguish between the features of protective perturbations and the original speech. For example, low-pass, band-pass filtering, and downsampling can remove certain frequency components of the speech, thereby affecting the ability of VC models to capture speaker characteristics. Additionally, the underperformance of the DualPure (Tan et al., 2024) and DiffSpec (Wu et al., 2023) can be attributed to their mel-spectrogram processing parameters. These methods operate on spectrograms with 32 mel-bins, which were initially designed for classification tasks. While 32 mel bins are sufficient for classification (for the classification models are trained on inputs of the same dimensionality), reconstructing waveforms from such a low-resolution spectrogram can lead to significant distortion. However, existing voice cloning models (Casanova et al., 2022; Betker, 2023) typically utilize mel-spectrograms with 80 mel bins or more as input, relying on high-fidelity spectral features. In contrast, WavePurifier (Guo et al., 2024) employs a higher resolution ($256 \times 256$) spectral representation, which preserves the essential details for voice cloning, thereby performing significantly better in our experiments. DiffWave, operat-

*Table 6.* Ablation study on the impact of data augmentation in different protection methods.

| Method | Purif. Stage | Ref. Stage | Data Aug. | Phon. Guide | AntiFake xSVA | AntiFake dSVA | AttackVC xSVA | AttackVC dSVA | VoiceGuard xSVA | VoiceGuard dSVA | Total xSVA | Total dSVA |
|---|---|---|---|---|---|---|---|---|---|---|---|---|
| WavePurifier | ✓ | N/A | N/A | N/A | 0.299 | 0.293 | 0.536 | 0.505 | 0.423 | 0.385 | 0.419 | 0.394 |
| AudioPure | ✓ | N/A | N/A | N/A | 0.401 | 0.451 | 0.734 | 0.776 | 0.656 | 0.712 | 0.597 | 0.646 |
| Ours (w/o Augmentation) | ✓ | ✓ | ✗ | ✓ | 0.582 | 0.689 | 0.747 | **0.872** | 0.716 | 0.812 | 0.682 | 0.791 |
| Ours (Full model) | ✓ | ✓ | ✓ | ✓ | **0.660** | **0.762** | **0.750** | 0.861 | **0.723** | **0.830** | **0.711** | **0.818** |

*Table 7.* Ablation study on the impact of speaker gender.

| Speaker Gender | Protected | WavePurifier | AudioPure | **Ours** |
|---|---|---|---|---|
| Female (16 speakers) | 0.103 | 0.383 | 0.603 | **0.796** |
| Male (9 speakers) | 0.106 | 0.414 | 0.677 | **0.856** |

ing directly in the waveform domain, avoids this type of distortion, similarly leading to better results.

### B.2. Quality of Synthesized Speech

Figure 10 shows the objective MOS of the synthesized speech from our method and existing adversarial purification methods. The results show that our method achieves greater naturalness across different VC models, indicating that our method not only improves the ability of VC models to replicate the target speaker's identity characteristics but also enhances their ability to synthesize natural speech. Numerical results are provided in Table 3.

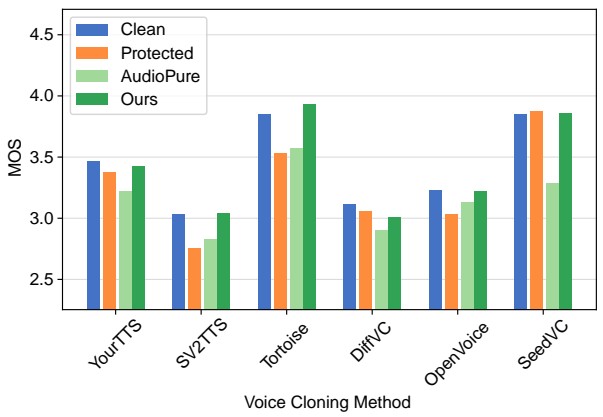

*Figure 10.* Objective MOS of VC models synthesized speech.

### B.3. Extended Ablation Study

**Impact of Refinement Steps.** Figure 11 shows the impact of different Refinement steps $N$ on the SVA of synthesized speech. We find that the Refinement model approaches optimal performance at step $N = 15$, and the SVA remains stable between steps 25 and 50. We choose $N = 30$ as the number of steps for the Refinement model in our experiments to balance performance and computational efficiency.

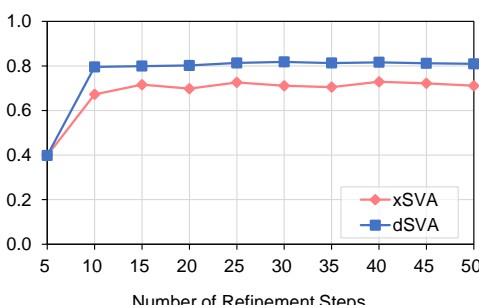

*Figure 11.* Impact of different Refinement steps.

*Table 8.* Cross-lingual performance of the Refinement stage.

| Metric | Protected | w/o Refinement | **Full Model** |
|---|---|---|---|
| dSVA | 0.212 | 0.933 | **0.981** |

**Impact of Data Augmentation.** We described our data augmentation strategy in App. A.5. The impact of data augmentation on SVA is presented in Table 6. The results indicate that applying data augmentation during the Refinement stage can slightly improve SVA in most cases. Furthermore, even without data augmentation, our method consistently outperforms existing purification methods, showing our method does not rely on data augmentation.

**Impact of Speaker Gender.** We present the dSVA results of our method and existing purification methods for speakers of different genders in Table 7. We observed that our method outperforms existing purification methods across both male and female speakers, achieving higher dSVA values. Additionally, all purification methods yield higher dSVA results for male speakers compared to female speakers. This may be because female voices typically contain more high-frequency information, making the separation of speech from high-frequency artifacts introduced by protective perturbations more challenging.

**Cross-Lingual Performance of the Refinement Stage.** Since phoneme representations are language-dependent, we investigate the impact of language on our method. We conduct a small-batch inference on the Russian LibriSpeech[4],

---
[4]https://www.openslr.org/96/

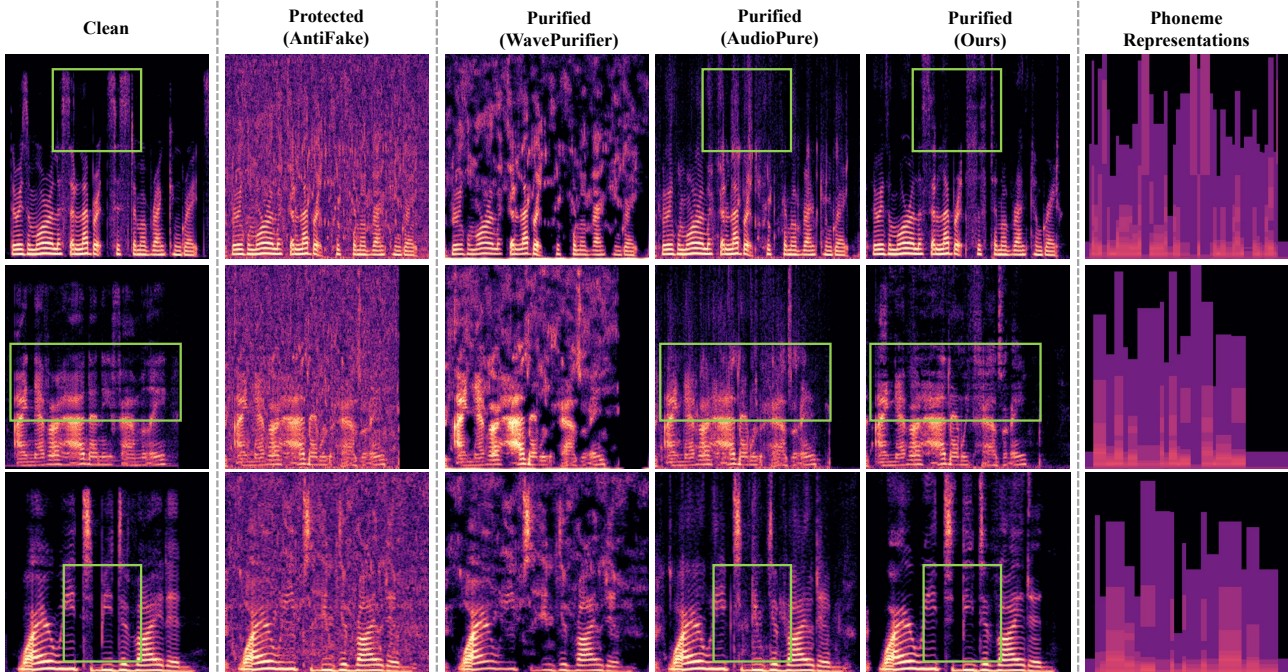

Figure 12. Spectrogram comparison of speech samples as voice cloning inputs. Existing purification methods tend to produce samples with similar patterns or blurred details, whereas our method retains details similar to those of the clean samples.

which utilizes a non-Latin script. We apply the AttackVC protection against the DiffVC model and use dSVA as the evaluation metric. Results in Table 8 indicate our Refinement stage remains effective even when inferring on a language distinct from the English training data.

### B.4. Visualized Results

The visualization of the samples as voice cloning inputs and outputs are shown in Figure 12 and Figure 13. As shown in Figure 12, existing purification methods tend to produce similar patterns. For example, WavePurifier introduces a blotchy pattern, while AudioPure's purified samples typically exhibit vertical stripes with blurred details, causing them to deviate from the clean distribution and leading to a decline in the performance of the VC models (see Figure 13). In contrast, samples purified with our method retain details similar to those of the clean samples, enabling the VC model to better capture the speaker's characteristics. These visual results align with both our design objectives and the numerical experiments.

### B.5. Time Cost

We run different purification methods on the NVIDIA RTX A6000 and obtain the average time spent processing each second of audio for each method, as shown in Table 9. Since our method introduces a score-based diffusion model during the Refinement phase, the computational time for our

Table 9. Average time consumption per second of audio sample. $N$ represents the number of Refinement steps.

| Method | WavePurifier | AudioPure | Ours ($N = 30$) | Ours ($N = 15$) |
|---|---|---|---|---|
| Time Cost (s) | 4.405 | 0.152 | 1.405 | 0.866 |

method is greater than that of AudioPure.

We believe this is acceptable for determined voice cloning attackers, as voice cloning attacks are typically performed offline. Attackers usually have only a few protected audio samples (ranging from a few seconds to a few minutes) and ample time to prepare their attacks. Moreover, our method's computational cost of our method can be reduced by decreasing the number of steps in the Refinement phase. For example, when the number of Refinement steps is set to $N = 15$, the SVA approaches optimal performance (as shown in Figure 11), and the processing time per second of speech sample is 0.866.

## C. Discussion

**Advanced Adaptive Protection Methods.** Although in section 5.4, white-box adaptive strategies like BPDA and adjoint struggle to generate effective perturbations in our experiments, several works in vision tasks (Kang et al., 2024; Lee & Kim, 2023b) have successfully performed adaptive adversarial attacks against diffusion-based adversarial pu-

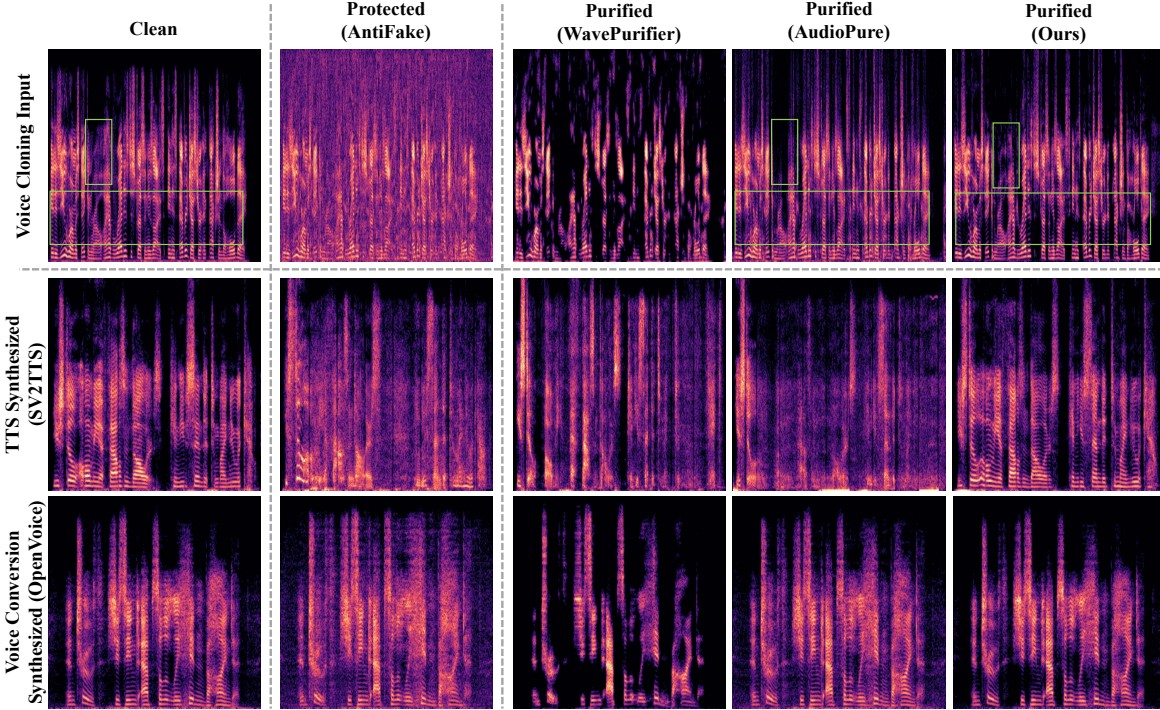

*Figure 13.* Visualization of voice cloning inputs and corresponding synthesized speech. The first row displays the speech samples as voice cloning input, while the second and third rows show the corresponding synthesized speech using different voice cloning models. Our method retains more details in the purified samples, thereby enhancing the VC model's ability to replicate the speaker's characteristics.

rification methods. As these methods are primarily designed for image classification tasks and face challenges such as increased memory usage due to the high resolution of speech data (*e.g.*, 96k data points for 6s audio at 16kHz vs. 1k for image in CIFAR-10), they cannot be directly applied to construct more robust protective perturbations against voice cloning attacks. However, by processing long audio in chunks to reduce memory overhead, or by using accelerated sampling methods as surrogates to reduce computational graph depth, their core mechanisms could potentially be adapted for speech tasks, enabling the construction of adaptive protection methods robust against our purification.

**Future Work on Protection Methods.** Our experimental results indicate that existing perturbation-based VC defenses are vulnerable when adversarial purification is present in the attack pipeline. Future work could explore more robust protective perturbation strategies, such as integrating distortion layers composed of different types of distortions into the perturbation generation pipeline to enhance their robustness against adversarial purification. Additionally, adding protective perturbations at more advanced speech feature levels may also enhance the robustness of protective perturbations.

**Discussion of Our Purification Method.** Although our purification method has outperformed existing methods

in experiments, there are still some limitations. For instance, our method employs fixed Purification and Refinement timestep in all experiments. However, some studies have shown that there exists an optimal timestep when diffusion models are used for adversarial purification (Guo et al., 2024; Nie et al., 2022). Therefore, a fixed timestep may limit the effectiveness of our method. Future work could explore the use of adaptive timestep selection strategies to improve purification effectiveness.

Additionally, our method does not utilize information from the VC model, which means that our method does not depend on a specific VC model and attacker's knowledge about it. However, this design leaves room for improvement: attackers with access to the VC model and expertise in it could exploit this information to increase the success rate of their attacks. For example, attackers could adversarially fine-tune the purification model using adversarial samples targeting their VC model to enhance its ability to purify adversarial perturbations. Future work could investigate how to leverage information from the VC model to improve purification effectiveness and thereby increase the success rate of VC attacks. Furthermore, although we have explored the use of a specific backbone as the Purification and Refinement model in this paper, our Purification-Refinement framework could benefit from more advanced Purification and Refinement models.

