# OpenReview forum: "De-AntiFake: Rethinking the Protective Perturbations Against Voice Cloning Attacks"
_ICML.cc/2025/Conference — ICML 2025 poster_

### Official Review · Reviewer_bzRG · 2025-03-11

**Overall Recommendation:** 4

**Summary:**

The paper investigates the effectiveness of adversarial perturbations as a defense against voice cloning (VC) under threat models that specifically considering perturbation purification techniques used by attackers. The study finds that while existing purification methods can reduce the impact of protective perturbations, they still cause distortions in the feature space of VC models, leading to a decline in voice cloning performance. Therefore, the paper proposes a novel two-stage purification method that first purifies the perturbed speech and then refines it using phoneme guidance to better align it with clean speech. The experimental results demonstrate that this new method outperforms existing purification techniques in disrupting voice cloning defenses. Ablation experiments validate the effectiveness of each component. Adaptive protection experiments demonstrate that the proposed method exhibits a degree of robustness.

**Claims And Evidence:**

The main claims of the paper include: (1) existing adversarial perturbations as a defense against voice cloning are vulnerable to existing adversarial purification; (2) the proposed new purification method outperforms existing methods in disrupting voice cloning defenses. Claims (1) and (2) are supported by subjective and objective experiments in Section 5.2, and the ablation study in Section 5.3 verifies the effectiveness of each component of the method in claim (2). In summary, the main claims of the paper are supported by clear and convincing evidence.

**Essential References Not Discussed:**

No.

**Experimental Designs Or Analyses:**

The experimental designs were checked for soundness and validity. Specifically, the designs in Sections 5.2 (main results), 5.3 (ablation study), and 5.4 (adaptive protection experiments) were checked. The designs are appropriate for testing the paper's claims, employing suitable datasets, metrics, and comparisons to relevant baselines.  No issues were found regarding the soundness or validity of the experimental designs or analyses.

**Methods And Evaluation Criteria:**

The proposed methods are make sense for addressing the problem.  The evaluation criteria, including the chosen metrics and baselines (defense methods and purification baselines), are in line with established practices and influential prior work in the field, making the evaluation robust and relevant.

**Other Comments Or Suggestions:**

While not strictly necessary for publication, open-sourcing the code would greatly enhance the reproducibility of this work and facilitate further research in this area.

**Other Strengths And Weaknesses:**

Strengths:
1. The paper highlights the vulnerability of existing perturbation-based voice cloning defenses in the presence of adversarial purification. This is a critical security concern for current voice cloning defenses.
2. The proposed new purification method is a creative combination of existing techniques and demonstrates some originality. Experimental results demonstrate its effectiveness.
3. The paper is well-organized, and the figures and tables are clear and informative.

Weaknesses:
1. The proposed method increases time cost due to the introduction of the refinement stage.
2. In Section 4.3, "Phoneme Representation," the terms "force alignment" and "force aligner" may require clarification. This lack of clarity makes it slightly challenging for readers unfamiliar with this content to fully grasp the method's implementation.
3. Some experimental results and settings require explanation (see the "Questions For Authors" section).

**Questions For Authors:**

1. The paper analyzes the limitations of existing purification methods, such as DiffWave from [2] and WavePurifier[3]. However, the paper does not analyze why DiffSpec from [2] or DualPure[4] performs worse than DiffWave or WavePurifier[3] in this task (as shown in Table 4). This is somewhat counterintuitive, as in other adversarial purification tasks, their performance difference compared to DiffWave is much smaller [2, 4].  A reasonable analysis of these results could significantly strengthen the paper. Could the authors provide an analysis of these seemingly unexpected results?
2. The paper compares the proposed method with "DS," "QT," and "Mel" methods from [1]. However, it does not include comparisons with the "Filter Power" and "LPC" methods also presented in [1]. Could the authors please either supplement the experiments to include these comparisons or provide a justification for why "Filter Power" and "LPC" methods were not included in the comparative evaluation?
3. In Table 4 (Appendix B.4), the "Parameter Value" for "DS" (Downsampling) is 1600Hz. For original audio with a sampling rate of 16000Hz, this parameter choice seems unusual for downsampling as it would lead to a significant loss of high-frequency components. The parameter also deviates from the default settings in Paper [1].  Could the authors please explain the rationale behind selecting this specific parameter value for downsampling?



[1] Hussain, S., Neekhara, P., Dubnov, S., McAuley, J., and Koushanfar, F. WaveGuard: Understanding and Mitigating Audio Adversarial Examples. USENIX, 2021.
[2] Wu, S., Wang, J., Ping, W., Nie, W., and Xiao, C. Defending against adversarial audio via diffusion model. In The Eleventh International Conference on Learning Representations, 2023.
[3] Guo, H., Wang, G., Chen, B., Wang, Y., Zhang, X., Chen, X., Yan, Q., and Xiao, L. WavePurifier: purifying audio adversarial examples via hierarchical diffusion models. In Proceedings of the 30th Annual International Conference on Mobile Computing and Networking, pp. 1268–1282, 2024.
[4] Tan, H., Liu, X., Zhang, H., Zhang, J., Qian, Y., and Gu, Z. DualPure: An Efficient Adversarial Purification Method for Speech Command Recognition. In Interspeech 2024, pp. 1280–1284. ISCA, September 2024.

**Relation To Broader Scientific Literature:**

The paper relates to the broader literature of using adversarial perturbations for voice cloning defense, as seen in works like [1-3].  The key contribution is in empirically investigating the impact of adversarial purification on these defenses in the voice cloning domain, providing valuable practical insights. While the proposed two-stage purification method combines existing techniques such as [4-6], the proposed method represents a useful improvement. The paper contributes to the field by providing a more nuanced understanding of the interplay between adversarial perturbations and purification in voice cloning, and suggests a direction for enhancing purification methods in this specific context.

[1] Yu, Z., Zhai, S., and Zhang, N. AntiFake: Using Adversarial Audio to Prevent Unauthorized Speech Synthesis. In Proceedings of the 2023 ACM SIGSAC Conference on Computer and Communications Security, pp. 460-474, Copenhagen Denmark, November 2023.
[2] Huang, C.-y., Lin, Y. Y., Lee, H.-y., and Lee, L.-s. Defending Your Voice: Adversarial Attack on Voice Conversion. In 2021 IEEE Spoken Language Technology Workshop (SLT), pp. 552-559, Shenzhen, China, January 2021.
[3] Li, J., Ye, D., Tang, L., Chen, C., and Hu, S. Voice Guard: Protecting Voice Privacy with Strong and Imperceptible Adversarial Perturbation in the Time Domain. In Proceedings of the Thirty-Second International Joint Conference on Artificial Intelligence, pp. 4812-4820, Macau, SAR China, August 2023b.
[4] Wu, S., Wang, J., Ping, W., Nie, W., and Xiao, C. Defending against adversarial audio via diffusion model. In The Eleventh International Conference on Learning Representations, 2023.
[5] Song, Y., Sohl-Dickstein, J., Kingma, D. P., Kumar, A., Ermon, S., and Poole, B. Score-based generative modeling through stochastic differential equations. In International Conference on Learning Representations, 2021.
[6] Tian, Y., Liu, W., and Lee, T. Diffusion-Based Mel-Spectrogram Enhancement for Personalized Speech Synthesis with Found Data. In 2023 IEEE Automatic Speech Recognition and Understanding Workshop (ASRU), pp. 1-7, December 2023.

**Theoretical Claims:**

This paper is primarily focused on empirical evaluation and algorithmic innovation in the domain of voice cloning defense.  It does not present formal theoretical claims or mathematical proofs in the traditional sense. Therefore, there were no proofs to check for correctness, and consequently, there are no issues related to proof validity to discuss.

---

> ### Author Rebuttal · Authors · 2025-03-31
>
> We thank the reviewer for the insightful and valuable comments. We respond to each comment as follows and sincerely hope that our rebuttal could properly address your concerns.
> # Weaknesses
> >W1. The proposed method increases time cost due to the introduction of the Refinement stage.
>
> We agree that the Refinement stage increases computational time compared to single-stage purification. However, voice cloning attacks are fundamentally offline processes. Attackers typically have **hours/days** to prepare critical audio samples (e.g., CEO's voice for fraud), making increased computation, which is **seconds-level** as shown in Table 8 still acceptable for these scenarios. For instance, an audio sample of 15 seconds only needs 13 seconds for our method to process.
> >W2. The terms "force alignment" and "force aligner" may require clarification.
>
> - The Force Aligner mentioned in our paper refers to the Montreal Forced Aligner (https://github.com/MontrealCorpusTools/Montreal-Forced-Aligner), which is a **tool** used for aligning text with audio. Specifically, it works by first converting the text into a sequence of phonemes. Then, using a pre-trained speech recognition model, it aligns the start and end times of each phoneme to precise timestamps within the corresponding audio.
>
> - Force alignment, on the other hand, refers to the **process** of performing this audio and text alignment using the Montreal Forced Aligner.
>
> To prevent any confusion, we will provide explicit definitions for both "force aligner" and "force alignment" in Section 4.3 of the revised paper.
> # Questions (& Weakness 3)
> >Q1. Please analyze why DiffSpec and DualPure perform unexpectedly worse than DiffWave and WavePurifier in your task compared to other purification tasks.
>
> The relatively poorer performance of DiffSpec [2] and DualPure [4] can be attributed to their mel-spectrogram processing parameters. These methods operate on **spectrograms with 32 mel-bins**, which were initially designed for classification tasks. While 32 mel bins are sufficient for classification (for the classification models are trained on inputs of the same dimensionality), reconstructing waveforms from such a low-resolution spectrogram can lead to significant distortion.
>
> However, existing voice cloning models [7, 8] typically utilize **mel-spectrograms with 80 mel bins or more** as input, relying on high-fidelity spectral features. In contrast, WavePurifier [3] employs a higher resolution (256x256) spectral representation, which preserves the essential details for voice cloning, thereby performing significantly better than DiffSpec [2] and DualPure [4] in our experiments. DiffWave, operating directly in the waveform domain, avoids this type of distortion, similarly leading to better results.
>
> We will include this analysis in Section 5.2 or the Appendix of the revised paper.
> >Q2. Omitted comparisons with the "Filter Power" and "LPC" methods.
>
> Filter Power and LPC from [1] were omitted in our initial experiments due to:
> - Filter Power functionality overlaps with BPF in [5] (both are band pass filter methods),
> - LPC functionality overlaps with MP3 in [5] (both are compression methods).
>
> We acknowledge the importance of direct comparison and conducted new experiments. As shown below, **both methods underperform our proposed approach** (results to be added to Table 5):
>
> Method|xSVA|dSVA
> -|-|-
> Filter Power| 0.222|0.133
> LPC|0.121|0.026
> Ours|0.711|0.818
>
> >Q3. The "Parameter Value" for Downsampling (1600Hz) needs explaination.
>
> The "1600Hz" entry in Table 4 is a typo. The correct downsampling rate is **8000Hz**, aligning with common practices in prior voice cloning defenses and adversarial purification [2, 5, 6]. For completeness, we also evaluated [1]'s default parameter (6000Hz), achieving:
>
> Method|Downsampling rate|xSVA|dSVA
> -|-|-|-
> Down-Up Sampling|6000Hz|0.204|0.140
> Down-Up Sampling|8000Hz|0.168|0.139
> Ours|N/A|0.711|0.818
>
> **both of them underperform our proposed approach**. We will correct this typo in Table 4.
> # Other Comments or Suggestions
> >Open-sourcing the code would greatly enhance the reproducibility of this work and facilitate further research in this area.
>
> Yes, we will make our code publicly available once our paper is accepted.
>
> [1] WaveGuard: Understanding and Mitigating Audio Adversarial Examples. USENIX21.
>
> [2] Defending against adversarial audio via diffusion model. ICLR23.
>
> [3] WavePurifier: purifying audio adversarial examples via hierarchical diffusion models. MobiCom24.
>
> [4] DualPure: An Efficient Adversarial Purification Method for Speech Command Recognition. Interspeech24.
>
> [5] Towards Understanding and Mitigating Audio  Adversarial Examples for Speaker Recognition. TDSC22.
>
> [6] AntiFake: Using adversarial audio to prevent unauthorized speech synthesis. CCS23.
>
> [7] Yourtts: Towards zero-shot multi-speaker tts and zero-shot voice conversion for everyone. ICML22.
>
> [8] Better speech synthesis through scaling. arXiv preprint 23.

---

> > ### Comment · Reviewer_bzRG · 2025-04-04
> >
> > Thank you for clarifying the comments. I will keep the score.

---

> > > ### Author Response · Authors · 2025-04-04
> > >
> > > Thank you for confirming the clarification. We appreciate your valuable feedback and support.

---

### Official Review · Reviewer_uK8k · 2025-03-13

**Overall Recommendation:** 4

**Summary:**

This paper investigates the vulnerabilities of protective perturbation-based voice clone (VC) defenses, demonstrating that these defenses are susceptible to existing adversarial purification techniques. Additionally, the authors propose an enhanced two-stage adversarial purification method that mitigates embedding inconsistencies caused by current methods, thereby further exposing the weaknesses of these VC defenses. Even with full access to the gradient information of their purification model, the study underscores the urgent need for more advanced techniques to prevent unauthorized data usage in VC. Both objective and subjective evaluations confirm the superior performance of their purification method.

**Claims And Evidence:**

The claims made in this paper are verified by experimental results, which include both objective and subjective evaluations.

**Essential References Not Discussed:**

To my knowledge, important references are included in this paper.

**Experimental Designs Or Analyses:**

- The selected voice cloning method, protection methods, and adversarial purification baselines are based on recent work.
- The experimental designs in this paper are comprehensive, including both objective and subjective evaluations.

**Methods And Evaluation Criteria:**

The issue highlighted in this paper is important. The systematic evaluation of protective perturbation-based VC defenses serves as a warning to the community that such perturbations can be mitigated by adversarial purification methods. The proposed two-stage adversarial purification method further emphasizes these risks. Experimental results support their claim.

**Other Comments Or Suggestions:**

1. The description of the summarized contributions could be refined, for example, by directly highlighting the specific risks identified.
2. Typos: In the caption of Figure 2, "(b-c), Purified" should be corrected to "(b-c), purified."

**Other Strengths And Weaknesses:**

Strength:
1. The problem addressed in this paper is meaningful for society.
2. The motivation behind the two-stage purification method is well articulated.
3. The experiments are comprehensive, and the effectiveness of different components in their method is validated through an ablation study.
4. The threat model is well considered, including adaptive attacks, and the experimental results demonstrate that their purification method is difficult to mitigate even in a white-box scenario.

Weaknesses:

1. The process of purification, specifically unconditional diffusion, requires further explanation. For instance, Equations (3) and (4) should explicitly include X_{adv}
2. The training details are insufficiently clear. For example, the statement "The Purification model is a pretrained unconditional DiffWave model" contrasts with the context, which states, "Our method trains two models separately." The authors should clarify the detailed settings.
3. In Figure 11, the authors note that "Existing purification methods tend to produce samples with similar patterns and blurred details," but the similar patterns are difficult to discern. This section also requires further clarification.

**Questions For Authors:**

Refer to Weakness.

**Relation To Broader Scientific Literature:**

- Voice Clone Attacks
- Adversarial Examples
- Adversarial Purfication Methods

**Theoretical Claims:**

The problem addressed in this paper is practical, and there are no theoretical claims made.

---

> ### Author Rebuttal · Authors · 2025-03-30
>
> We thank the reviewer for the insightful and valuable comments. We respond to each comment as follows and sincerely hope that our rebuttal could properly address your concerns.
> # Weaknesses
> >W1. The process of purification, specifically unconditional diffusion, requires further explanation. For instance, Equations (3) and (4) should explicitly include $x_\text{adv}$.
>
> In Equations (3) and (4), $x_t$ refers to audio sample at time $t$, $x_\text{adv}$ refers to adversarial audio sample, and $t$ ranges from $1$ to $T_\text{pur}$. The foward process starts with $x_{0}$ ($x_\text{adv}$) and generates noisy sample $x_{T_\text{pur}}$; the reverse process starts with $x_{T_\text{pur}}$ and generates $x_{0}$ ($x_{\text{pur}}$). We will clarify these terms in Section 4.2 as follows:
> - (**Revision**) At each timestep $t$, the process is expressed as:
> $q(x_t | x_{t-1}) = \mathcal{N}(x_t; \sqrt{1-\beta_t} x_{t-1}, \beta_t \mathbf{I}), t = T_\text{pur}, T_\text{pur}-1, ..., 1$
> where $\beta_t$ denotes the variance schedule, and $x_0 = x_\text{adv}$.
> - The reverse process denoises the waveform $x_{T_\text{pur}}$ in the same $T_\text{pur}$ steps, and at each step, it is formulated as:
> $p_{\theta}(x_{t-1} | x_t) = \mathcal{N}(x_{t-1}; \mu_\theta(x_t, t), \sigma_t^2 \mathbf{I}),$
> where $\mu_\theta(x_t, t)$ is the mean function parameterized by $\theta$, $\sigma_t^2$ is the time-dependent variance schedule, and $x_0 = x_\text{pur}$.
>
> >W2. The training details are insufficiently clear. For example, the statement "The Purification model is a pretrained unconditional DiffWave model" contrasts with the context, which states, "Our method trains two models separately." The authors should clarify the detailed settings.
>
> Thank you for pointing out this ambiguity. We will clarify the description, avoiding the term "pretrained" to describe a model fine-tuned on a specific dataset as follows:
>
> - (**Revision**) The Purification model is **based on** a pretrained unconditional DiffWave model which is **then fine-tuned** on the LibriSpeech dataset.
>
> >W3. In Figure 11, the authors note that "Existing purification methods tend to produce samples with similar patterns and blurred details," but the similar patterns are difficult to discern. This section also requires further clarification.
>
> We apologize for not clearly explaining "the similar patterns." This was because the pattern is not evident from the small number of examples provided. We will address this by adding more examples in Figure 11 to clarify these patterns.
>
> # Other Comments or Suggestions
> >C1. The description of the summarized contributions could be refined, for example, by directly highlighting the specific risks identified.
>
> We will refine the descriptions of our contributions in the introduction by directly highlighting the specific risks identified. Details are as follows:
>
> - (**Revision**) We assess six VC methods and three protective techniques, revealing **the risk that existing defenses potentially fail to prevent voice cloning attacks**.
>
> >C2. Typos: In the caption of Figure 2, "(b-c), Purified" should be corrected to "(b-c), purified."
>
> Thank you for pointing out this typo. We will correct it in our revised paper.

---

### Official Review · Reviewer_JFuf · 2025-03-14

**Overall Recommendation:** 4

**Summary:**

The paper evaluates limitations of existing purification methods in countering adversarial perturbations designed to block unauthorized voice cloning (VC), revealing they cause feature distortions that degrade VC performance. A novel two-stage purification method is proposed, combining perturbation removal with phoneme-guided refinement to align purified speech with clean speech distribution. Experiments demonstrate this approach outperforms state-of-the-art methods in disrupting VC defenses, highlighting vulnerabilities in current adversarial perturbation-based security strategies.

**Claims And Evidence:**

The paper's claims are appropriate and well-supported. The authors claim that they are the first to explore vulnerabilities of protective VC (voice cloning) defenses, which is accurate and justified, as demonstrated through comprehensive experiments validating the proposed method's superiority. The substantial experimental evidence effectively reinforces the validity of their claims.

**Essential References Not Discussed:**

None

**Experimental Designs Or Analyses:**

The experimental and analyses of this paper are sound and valid. It compares with many adversarial purification baselines and analyzes the results on many tables and visualization figures.

**Methods And Evaluation Criteria:**

The method presented in this paper demonstrates important innovation, proposing a novel two-stage purification-refinement architecture. Additionally, the use of evaluation metrics such as xSVA, dSVA, and Mean Opinion Score (MOS) is well-justified and contextually appropriate for assessing the framework's effectiveness.

**Other Comments Or Suggestions:**

Please refer to the weakness.

**Other Strengths And Weaknesses:**

**Strength**

1. The writing of this paper is smooth and well-organized. The task scenario is clearly defined through the threat model and Figure 1, while the method is introduced in a general-to-specific manner, providing a logical and easy-to-follow structure. Additionally, the related works are presented in a clear and concise way.

2. The experiments are extensive and comprehensive, involving a wide range of voice cloning and protection methods. The comparison methods include most SOTA purification methods. Additionally, detailed ablation studies are provided to thoroughly analyze the impact of different components and parameters, further enhancing the robustness and credibility of the research.

**Weakness**

1. The purification method employed is relatively outdated, specifically an unconditional DiffWave model. Could the proposed method be generalized to other audio diffusion models?

2. Although the time complexity presented in Table 8 is acceptable, it is suggested that the authors consider adopting some diffusion model acceleration sampling techniques or flow-matching models to further improve the speed of purification. Additionally, I am curious about the respective time costs of the purification stage and the refinement stage. Please list them.

3. In Section 4.2, the sampling process of the diffusion model is not clearly articulated. It is recommended that the authors replace Equation 5 with an iterative sampling formula (such as DDPM or DDIM).

**Questions For Authors:**

None

**Relation To Broader Scientific Literature:**

None

**Theoretical Claims:**

The paper elaborates on the purification process in detail via Equations (3)-(5).

---

> ### Author Rebuttal · Authors · 2025-03-30
>
> We thank the reviewer for the insightful and valuable comments. We respond to each comment as follows and sincerely hope that our rebuttal could properly address your concerns.
> # Weaknesses
> >W1. The purification method employed is relatively outdated, specifically an unconditional DiffWave model. Could the proposed method be generalized to other audio diffusion models?
>
> Recent audio diffusion models are often designed for specific tasks such as conditional generation or speech enhancement, and therefore, the majority of them are conditional models. We experimented with applying newer conditional models [1, 2] to our current task, and the table below shows the dSVA after applying only the Purification stage with different models:
> | Purification Model | dSVA |
> | --- | --- |
> | Uncond. DiffWave | 0.647 |
> | SGMSE [1] | 0.638 |
> | DMSE [2] | 0.598 |
>
> We find that their performance was generally not as good as unconditional DiffWave. We infer that **an unconditional model is more suitable as our Purification model**, possibly because it lacks conditional constraints and might be more robust to unseen noise.
>
> Despite this, we attempted to apply another audio diffusion model to our task, specifically the one from [2], to investigate the potential of our method to generalize to other diffusion models. The resulting dSVA values are presented in the following table. The results indicate that when using another audio diffusion model as the Purification model, our Refinement stage also improved performance. This suggests that **our method has the potential to generalize to other audio diffusion models.**
> | Purification Model | dSVA (Purification Stage Only) | dSVA (Full Model) |
> | --- | --- | --- |
> | DMSE [2] | 0.598 | 0.622 |
>
> >W2. It is suggested that the authors consider adopting some diffusion model acceleration sampling techniques or flow-matching models to further improve the speed of purification. Additionally, I am curious about the respective time costs of the purification stage and the refinement stage. Please list them.
>
> - We list the processing time per second of audio for the Purification and Refinement stages separately in the table below. The results indicate that the Refinement stage takes up the majority of the processing time.
> | | Purification Stage | Refinement Stage (N=15) | Refinement Stage (N=30) |
> |----|----|----|----|
> | Time Cost (s) | 0.152 | 0.715 | 1.253 |
> - Your suggestion regarding acceleration is very constructive. We agree that acceleration sampling or flow-matching models could be helpful for real-world efficiency. Due to time constraints, we will explore potential accelerated sampling techniques or more efficient models in our future work to enhance the  practicality of our method for real-world applications and advance the field.
>
> >W3. In Section 4.2, the sampling process of the diffusion model is not clearly articulated. It is recommended that the authors replace Equation 5 with an iterative sampling formula (such as DDPM or DDIM).
>
> We will clarify the sampling process in Section 4.2, including replacing Equation 5 with the following iterative sampling formula:
>
> $x_{t-1} \sim p_{\theta}(x_{t-1} | x_t) = N(x_{t-1}; \mu_{\theta}(x_t, t), \sigma_t^2 {I}), t = T_\text{pur}, T_\text{pur}-1, ..., 1$
>
> where $x_0 = x_\text{pur}$.
>
> [1] Richter J, et al. Speech enhancement and dereverberation with diffusion-based generative models. IEEE/ACM Transactions on Audio, Speech, and Language Processing (TASLP), 2023.
>
> [2] Tian Y, Liu W, Lee T. Diffusion-Based Mel-Spectrogram Enhancement for Personalized Speech Synthesis with Found Data. IEEE Automatic Speech Recognition and Understanding Workshop (ASRU), 2023.

---

### Official Review · Reviewer_VwKh · 2025-03-14

**Overall Recommendation:** 3

**Summary:**

Some works claim that an individual can protect their audio samples from voice cloning via perturbations that induce odd behavior from the voice cloning model (generative), or from voice classification models (discriminative). Another set of works show that these defensive perturbations can be removed at least for classification models, effectively removing the defensive capabilities. But the authors contend that these methods also remove natural features and thereby induce a distribution shift from the original audio. This is apparently fine for discriminative models that don't need all the features, but not for generative models which do. So the voice cloning no longer works. The authors thus propose an additional phenome-based refining step, that essentially maps from the shifted distribution back to the original distribution.

**Claims And Evidence:**

- One should be careful about using the "human" notation of H. This is very hard to define. My human is different from yours. You eventually try some proxy evaluation using the embeddings to show reduced distortion using your method. But this doesn't necessarily imply reduced distortion to a human. A somewhat silly example would be me finding point using PGD in the embedding space that is within some ball of the clean audio, but when has no meaning when mapped back to the input space. I'm not sure how exactly you can change this. Simply adding some author evaluation would be fine I think.

**Essential References Not Discussed:**

NA

**Experimental Designs Or Analyses:**

- I am not convinced entirely by your adaptive protection discussion. I can preface this by saying I am not asking for more experiments, or that you are even mandated to explore these routes given the scope limits enforced by the page limits of an ML venue paper. But, you should discuss the adaptive attack works that have been effective against adversarial purification like DiffAttack (https://proceedings.neurips.cc/paper_files/paper/2023/file/ea0b28cbbd0cbc45ec4ac38e92da9cb2-Paper-Conference.pdf). Sure, it is the audio domain. But what would a defender need to do to run something like DiffAttack here, and make their perturbations survive? It seems rather intuitive to me that BPDA+EoT wouldn't work that well here since the papers you cite evaluated them and were also "robust" to them. Again, the onus is on the defenders here to make a better defense, but improved discussion would improve your paper.

**Methods And Evaluation Criteria:**

- Quite honestly Figure 2 does not easily communicate that there is reduced inter-class separability after applying diffusion-based-perturbation-removal. The points are too small, and there are too many speakers, i.e., colors. Can you also show that embedding results after applying only step 1 of your method? That should align with (b) and (c), and would communicate that you're doing +1 step on top of them.

**Other Comments Or Suggestions:**

- Can you please explain in the text how voice-cloning models work? At least the inference? Figure 1 is not explicit enough. I understand you use audio embeddings from the victim, and I am guessing you take text embeddings from the adversary? Or is that also audio embeddings? I suppose it's not critical to the paper but it's rather unfortunate that I went through the entirety of this paper and still don't have the remotest idea how voice cloning works.

**Other Strengths And Weaknesses:**

- Strengths:
    - The community is moving towards a consensus that protective perturbations do not work for the image domain, a la glaze. It is nice to show this for audio.
    - First to evaluate for cloning, which honestly is the more realistic threat, at least when compared to discriminative applications.

- Weaknesses/questions:
    - I think there is some assumption here (that likely holds true since your experiments work) that the two audio distributions induced by applying diffusion-based-perturbation-removal (your first step) to clean and defensively-perturbed audio are nearly the same. So, that would be why you can train your refinement model that maps back to the original distribution using some pairs from - (clean sample, diffusion-based-perturbation-removal(clean sample)). And then you can apply this at inference time to some diffusion-based-perturbation-removal(defensively-perturbed point), and expect to obtain a clean sample.
  - It's not very intuitive to me that the diffusion-based-perturbation-removal always outputs the same or 	nearly same audio distribution regardless of the input distribution being clean or adversarial. That in and of itself is an observation, so i think you should mention this at least somewhere. If it were not true you would need to run these attacks to generate training data for refinement, and that seems more likely to fail to generalize to unseen attacks.

**Questions For Authors:**

NA

**Relation To Broader Scientific Literature:**

The paper falls in line with attacks against perturbation-based training disruption defenses in other domains like face recognition and style transfer.

**Theoretical Claims:**

NA

---

> ### Author Rebuttal · Authors · 2025-03-30
>
> We thank the reviewer for the insightful and valuable comments. We respond to each comment as follows and sincerely hope that our rebuttal could properly address your concerns.
> # Claims And Evidence
> >Concern about subjective 'human' notation (H). Embedding proxy may not reflect human perception. Suggest adding author evaluation.
>
> - We follow [1] using similar notation $H$. We will make this clearer in the paper: $H(\cdot)$ represents the *perceived speaker identity* evaluated by human listeners with given audio.
> - To clarify, (clean, protected, purified) speech is the *reference speech* (VC model input). *Synthesized speech* is the VC model output, potentially used to deceive humans/SV models. We used the VC model's embedding as a proxy evaluation for 'purified speech' as it's the VC model's input; our goal is reduced distortion for the cloning model in 'purified speech' to facilitate successful voice cloning (deceiving SV models/humans). Humans don't evaluate 'purified speech'. However, synthesized speech (VC model output from source speech) may deceive humans.
> - Therefore, we have a subjective test (Sec. 5.2) with 20 humans evaluating *synthesized speech* from (clean, protected, purified) inputs, aligning with our threat model's $H$. For the purified speech, we don't involve human evaluation in the threat model (Sec. 3), so we think it is enough to use VC model embedding as proxy evaluation.
> # Methods And Evaluation Criteria
> >Fig. 2 unclear on reduced inter-class separability. Show embeddings after step 1 and full 2-steps for comparison.
> - We add some arrows pointing from clean to purified samples within Fig. 2(b-c) of anonymous link https://github.com/de-antifake/1/blob/main/2.png to better convey the reduced inter-class separability.
> - Fig. 2(c) already shows the embedding results after applying only step 1 (our method uses DiffWave for Purification stage, which is consistent with AudioPure), while Fig. 2(d) shows the results of applying the complete 2-step method.
> # Experimental Designs Or Analyses
> >Discuss adaptive attack against purification like DiffAttack in the audio domain.
> - We actively experimented with DiffAttack's core mechanisms (deviated-reconstruction loss and segment-wise forwarding-backwarding) following public code on an RTX A6000 (48GB VRAM). However, audio data's high resolution (e.g., 96k data points for 6s audio at 16kHz vs. 1k for DiffAttack's CIFAR-10) caused out-of-memory issues for DiffAttack on our task.
> - Possible solutions include using a lower-resolution diffusion model as surrogate, or processing long audio in chunks to reduce memory overhead. Furthermore, using accelerated sampling methods as surrogate to reduce computational graph depth might also help. Overall, running something like DiffAttack in the audio domain is challenging for defenders due to the high cost of calculating diffusion model gradients. Due to time, we didn't do further experiments, but future work can explore these adaptive defense methods. We will add this discussion to Sec. 5.4 or App. D in the revised paper.
> # Weaknesses/Questions
> >W1. & W2. There is an assumption: diffusion-based removal leads to similar distributions for clean & protected audio in the paper, which allows training refinement on (clean, purified(clean)) and applying to purified(protected). This non-intuitive observation should be mentioned in the paper.
>
> We show embedding distributions of Clean vs Protected, and Purified(Clean) vs Purified(Protected) Speech in anonymous link https://github.com/de-antifake/1/blob/main/3.png. We observed that diffusion-based perturbation removal brings clean and protected audio to similar distributions, which is a key assumption for our method's effectiveness. Thus, we don't need to generate adversarial data for training. We will mention this observation in the revised paper to clarify the Refinement stage's motivation.
> # Other Comments or Suggestions
> >Explain voice cloning model inference in the text.
>
> We apologize for not providing a clear explanation of how VC models work. Our work considers two main types of voice cloning: Text-to-Speech (TTS) and voice conversion. Both leverage speaker embeddings extracted from the victim's audio to capture their voice characteristics.
> - For TTS-based voice cloning, the attacker provides arbitrary text. The TTS acoustic model, conditioned on the victim's speaker embeddings, takes this text and generates corresponding acoustic features (e.g., mel-spectrograms). These acoustic features are then used by the vocoder to synthesize the cloned voice.
> - For voice conversion, the model typically takes the acoustic features of an arbitrary speaker's utterance provided by attacker (representing the content) and transforms them using the victim's speaker embeddings to generate speech with the victim's voice.
>
> We will provide a detailed explanation of these mechanisms in the revised paper.
>
> [1] Antifake: Using adversarial audio to prevent unauthorized speech synthesis. CCS23.

---

### Official Review · Reviewer_JTkR · 2025-03-15

**Overall Recommendation:** 3

**Summary:**

The paper "De-AntiFake" systematically evaluates current voice cloning defense mechanisms that use protective perturbations, revealing their vulnerability to adversarial purification techniques. To demonstrate this vulnerability, the authors propose a novel two-stage purification method called PhonePuRe that combines unconditional diffusion with phoneme-guided refinement to effectively bypass these protections. Through extensive experiments across six voice cloning methods and three protection techniques, they show their method significantly outperforms existing purification approaches, achieving higher speaker verification accuracy and better perceptual similarity even against adaptive protections.

**Claims And Evidence:**

The claims are generally well-supported with comprehensive experimental evidence. The authors thoroughly evaluate three protection methods (AntiFake, AttackVC, VoiceGuard) against six VC models, showing convincingly that their method achieves higher speaker verification accuracy.

**Essential References Not Discussed:**

Not aware.

**Experimental Designs Or Analyses:**

The experimental design is sound:

The evaluation dataset is appropriate and of sufficient size. The comparison with five existing adversarial purification methods is comprehensive. And the discussion of adaptive defense protection is very helpful.

**Methods And Evaluation Criteria:**

The methods and evaluation criteria are appropriate for the problem. The speaker verification accuracy (SVA) metrics (xSVA and dSVA) are standard and appropriate measures for evaluating the effectiveness of voice cloning and protection bypassing. The objective MOS score and human evaluation of perceived speaker similarity provide complementary evidence for perceptual assessment.

**Other Comments Or Suggestions:**

- Explore Trade-offs Between Attack Success and Audio Quality: Consider a more detailed analysis of the trade-offs between successful bypassing of protections and the resulting audio quality.

**Other Strengths And Weaknesses:**

- Ablation study like the discussion of adaptive protection is very well-conducted.

Weaknesses:

- The paper could benefit from more discussion about the practical limitations of their attack (e.g., computational requirements for real-world deployment)
- The author should consider include some theoretical discussion which can give a better understanding on the limitations of this methods.
- The paper could explore more deeply whether there are fundamental limitations to protective perturbation approaches given the effectiveness of their purification method
- The author should define dSVA and xSVA with explicit math formulation.

**Questions For Authors:**

- How does your PhonePuRe method perform when dealing with speech in different languages outside of the training data? Since phoneme representations are language-dependent, would your refinement stage maintain its effectiveness for languages with significantly different phonetic structures?
- The paper demonstrates the vulnerability of current protective perturbation methods, but what do you believe are the most promising directions for creating more robust voice cloning defenses that could withstand purification attacks like yours?
- Your time cost analysis in Appendix C.5 shows that PhonePuRe requires more computational resources than some existing methods. Have you explored potential optimizations that could reduce this overhead while maintaining similar performance?
- Your experiments focus on zero-shot voice cloning models. Would your conclusions and the effectiveness of your method change when considering few-shot or many-shot voice cloning attacks where attackers have access to more reference audio?
- The paper mentions that your method performs better for male speakers than female speakers. Could you elaborate on the underlying technical reasons for this gender disparity and potential approaches to address it?
- Given that your method significantly reduces the effectiveness of current protective perturbations, do you think there is a fundamental limitation to adversarial perturbation-based defenses against voice cloning, or is this an arms race that will continue to evolve?

**Relation To Broader Scientific Literature:**

I think this work points a good direction for the community where people focused on the protective perturbations.
Not aware.

**Theoretical Claims:**

The author should consider developing theory for why their proposed is able to perform effective VC against various protection. E.g. certain transformation/diffusion computation can mitigate the bounded adversarial noise. It would be helpful to know to what extent (protective setup) we can view this 2-stage procedure as effective. The condition could be sth like the signal is not degraded too much from the transformation

---

> ### Author Rebuttal · Authors · 2025-03-30
>
> We thank the reviewer for the insightful and valuable comments. We respond to each comment as follows and sincerely hope that our rebuttal could properly address your concerns.
> # Weaknesses
> >W1. More discussion about practical limitations.
>
> Our implementation utilized an RTX A6000 GPU and Xeon Gold 6130 @2.1GHz CPU. The peak computational resource usage is detailed below:
> ||GPU RAM|CPU RAM|GPU Usage|CPU Usage
> -|-|-|-|-
> Train|25.7G|6.0G|100%|433%
> Inference|9.2G|5.2G|100%|124%
>
> These requirements indicate that executing our attack is **not feasible on lightweight devices, and requires moderate computational resources** as illustrated in our threat model (Sec. 3).
> >W2. **(& Theoretical Claims)** Consider developing theory for better understanding of proposed method's interpretability and limitations.
>
> Thank you for your insightful suggestion. According to the proof in [1], as the forward diffusion time $t$ increases, the KL divergence between the clean distribution $p_t$ and the adversarial distribution $q_t$ is monotonically decreasing, meaning the clean and the adversarial distribution become closer. This provides theoretical support for our Purification stage to effectively mitigate adversarial noise. Building upon this, developing a systematic theory specific to our 2-stage method will lead to a better understanding of our method's limitations and interpretability. However, due to time/space limitations, we can only explore this issue in future work.
> >W3. **(& Question 6)** Do you think there is a fundamental limitation to protective perturbations?
>
> Though our method shows limitations in existing VC defenses, we did not 100% success in bypassing the protections, indicating that current defenses still offer some level of protection. Furthermore, as is common in all areas of information security research, defenders will likely develop countermeasures against our purification. Therefore, we believe **it is premature to assert a fundamental limitation of existing protections, and the arms race between attack and defense will likely continue.**
> >W4. Define dSVA and xSVA with math formulation.
>
> We will add definitions for SVA like follows: $dSVA = \frac{1}{N} \sum_{i=1}^{N} SV_d(x_{test}^i, x_{clean}^i)$.
>
> # Questions
> >Q1. Cross-lingual performance of the Refinement stage.
>
> We show the results of a small-batch inference on Russian LibriSpeech (non-Latin script) below. Our Refinement stage **remains effective** outside the training data (English).
> | |Protected|w/o Refinement|Full Model
> -|-|-|-
> dSVA|0.21|0.93|0.98
> >Q2. Future directions for robust VC defenses.
>
> We think embedding protective perturbations in higher-level semantic features for robustness appears promising. Combining perturbations with other defenses like proactive watermarking for a multi-layered defense is another promising direction.
> >Q3. Optimizations for computational overhead.
>
> Our paper focuses on revealing existing defense vulnerabilities, and we believe the current time cost (second-level) is acceptable for offline attacks. Your question is constructive, due to time, we will strive to optimize in future work to enhance real-world efficiency.
> >Q4. Effectiveness for few/many-shot VC attacks.
>
> Although existing VC defenses [2-4] don't consider few/many-shot scenarios, we run a small-batch VC attack where attacker have 5 reference audio and find our method **remains effective**:
> | |Protected|AudioPure|Ours
> -|-|-|-
> dSVA|0.03|0.17|**0.82**
>
> >Q5. Reasons and potential solutions for gender disparity.
>
> Figure in anonymous link https://github.com/de-antifake/1/blob/main/1.png show protective perturbations mainly affect higher frequencies where female voices have more energy. We infer **the frequency overlap makes separating perturbations from female voice characteristics harder** during purification, explaining the disparity. Potential solutions include hierarchical purification: use different purification steps for high/low frequencies could mitigate this.
>
> # Other Comments Or Suggestions
> > Explore trade-offs between attack success and audio quality.
>
> We **do not observe a clear trade-off** between audio quality and attack success. We use PESQ to evaluate the audio quality of the purified audio using a small batch of data:
> Purification Steps|1|2|3|4|5
> -|-|-|-|-|-
> PESQ|1.56|1.81|**1.88**|1.87|1.81
> dSVA|0.87|0.94|**0.96**|0.92|0.90
>
> As purification steps increases, both PESQ and dSVA initially improve and then decline, reaching their optimal values at the same step. This suggests that the protective noises degrades both audio quality and attack success, and proper number of purification steps improves both.
>
> [1] Diffusion Models for Adversarial Purification. ICML22.
>
> [2] AntiFake: Using Adversarial Audio to Prevent Unauthorized Speech Synthesis. CCS23.
>
> [3] Defending Your Voice: Adversarial Attack on Voice Conversion. SLT21.
>
> [4] Voice Guard: Protecting Voice Privacy with Strong and Imperceptible Adversarial Perturbation in the Time Domain. IJCAI23.

---

### Decision · Program_Chairs · 2025-05-01

**Decision:**

Accept (poster)

**Comment:**

Thank you for submitting your work to ICML 2025, and for your efforts in the rebuttal phase to clarify the reviewers' concerns. The authors should clarify the motivation behind the refinement stage, as it serves as a key assumption in the paper. Additionally, evaluating the proposed method across different languages would further support its effectiveness. I believe these issues can be effectively addressed in the camera-ready version. Since all the reviewers affirmed the novelty of the authors' work, I recommend accepting the paper.